# Understanding Deep Architectures with Reasoning Layer

**Xinshi Chen**
Georgia Institute of Technology
xinshi.chen@gatech.edu

**Yufei Zhang**
University of Oxford
yufei.zhang@maths.ox.ac.uk

**Christoph Reisinger**
University of Oxford
christoph.reisinger@maths.ox.ac.uk

**Le Song**
Georgia Institute of Technology
lsong@cc.gatech.edu

## Abstract

Recently, there has been a surge of interest in combining deep learning models with reasoning in order to handle more sophisticated learning tasks. In many cases, a reasoning task can be solved by an iterative algorithm. This algorithm is often unrolled, and used as a specialized layer in the deep architecture, which can be trained end-to-end with other neural components. Although such hybrid deep architectures have led to many empirical successes, the theoretical foundation of such architectures, especially the interplay between algorithm layers and other neural layers, remains largely unexplored. In this paper, we take an initial step towards an understanding of such hybrid deep architectures by showing that properties of the algorithm layers, such as convergence, stability and sensitivity, are intimately related to the approximation and generalization abilities of the end-to-end model. Furthermore, our analysis matches closely our experimental observations under various conditions, suggesting that our theory can provide useful guidelines for designing deep architectures with reasoning modules (i.e., algorithm layers).

## 1 Introduction

Many real world applications require perception and reasoning to work together to solve a problem. Perception refers to the ability to understand and represent inputs, while reasoning refers to the ability to follow prescribed steps and derive answers satisfying certain constraints. To tackle such sophisticated learning tasks, recently, there has been a surge of interests in combining deep perception models with reasoning modules (or algorithm layers).

Typically, a **reasoning module** is stacked on top of a **neural module**, and treated as an additional layer of the overall deep architecture; then all the parameters in the architecture are optimized end-to-end with loss gradients (Fig 1). Very often these reasoning modules can be implemented as unrolled *iterative algorithms*, which can solve more sophisticated tasks with carefully designed and interpretable operations. For instance, SATNet [1] integrated a satisfiability solver into its deep model as a reasoning module; E2Efold [2] used a constrained optimization algorithm on top of a neural energy network to predict and reason about RNA structures, while [3] used optimal transport algorithm as a reasoning module for learning to sort. Other algorithms such as ADMM [4, 5], Langevin dynamics [6], inductive logic programming [7], DP [8], k-means clustering [9], message passing [10, 11], power iterations [12] are

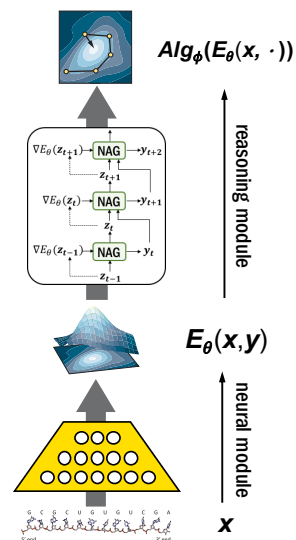

Figure 1: Hybrid architecture.

also used as differentiable reasoning modules in deep models for various learning tasks. Thus in the remainder of the paper, we will use reasoning module and algorithm layer interchangeably.

While these previous works have demonstrated the effectiveness of combining deep learning with reasoning, the theoretical underpinning of such hybrid deep architectures remains largely unexplored. For instance, what is the benefit of using a reasoning module based on unrolled algorithms compared to generic architectures such as recurrent neural networks (RNN)? How exactly will the reasoning module affect the generalization ability of the deep architecture? For different algorithms which can solve the same task, what are their differences when used as reasoning modules in deep models? Despite the rich literature on rigorous analysis of algorithm properties, there is a paucity of work leveraging these analyses to formally study the learning behavior of deep architectures containing algorithm layers. This motivates us to ask the crucial and timely question of

> *How will the algorithm properties of an algorithm layer affect the learning behavior*
> *of deep architectures containing such layers?*

In this paper, we provide a first step towards an answer to this question by analyzing the approximation and generalization abilities of such hybrid deep architectures. To the best our knowledge, such an analysis has not been done before and faces several difficulties: 1) The analysis of certain algorithm properties such as convergence can be complex by itself; 2) Models based on highly structured iterative algorithms have rarely been analyzed before; 3) The bound needs to be sharp enough to match empirical observations. In this new setting, the complexities of the algorithm's analysis and generalization analysis are intertwined together, making the analysis even more challenging.

**Summary of results.** We find that standard Rademacher complexity analysis, widely used for neural networks [13, 14, 15], is insufficient for explaining the behavior of these hybrid architectures. Thus we resort to a more refined local Rademacher complexity analysis [16, 17], and find the following:

- **Relation to algorithm properties.** Algorithm properties such as convergence, stability and sensitivity all play important roles in the generalization ability of the hybrid architecture. Generally speaking, an algorithm layer that is faster converging, more stable and less sensitive will be able to better approximate the joint perception and reasoning task, while at the same time generalize better.
- **Which algorithm?** There is a tradeoff that a faster converging algorithm has to be less stable [18]. Therefore, depending on the precise setting, the best choice of algorithm layer may be different. Our theorem reveals that when the neural module is over- or under-parameterized, stability of the algorithm layer can be more important than its convergence; but when the neural module is has an 'about-right' parameterization, a faster converging algorithm layer may give a better generalization.
- **What depth?** With deeper algorithm layers, the representation ability gets better, but the generalization becomes worse if the neural module is over/under-parameterized. Only when it has 'about-right' complexity, deeper algorithm layers can induce both better representation and generalization.
- **What if RNN?** It has been shown that RNN (or graph neural networks, GNN) can represent reasoning and iterative algorithms [19, 15]. On the example of RNN we demonstrate in Sec 6 that these generic reasoning modules can also be analyzed under our framework, revealing that RNN layers induce better representation but worse generalization compared to traditional algorithm layers.
- **Experiments.** We conduct empirical studies to validate our theory and show that it matches well with experimental observations under various conditions. These results suggest that our theory can provide useful practical guidelines for designing deep architectures with algorithm layers.

**Contributions and limitations.** To the best of our knowledge, this is the first result to quantitatively characterize the effects of algorithm properties on the learning behavior of hybrid deep architectures with reasoning modules (algorithm layers), showing that algorithm biases can help reduce sample complexity of such architectures. Our result also reveals a subtle and previously unknown interplay between algorithm convergence, stability and sensitivity when affecting model generalization, and thus provides design principles for deep architectures with algorithm layers. To simplify the analysis, our initial study is limited to a setting where the reasoning module is an unconstrained optimization algorithm and the neural module outputs a quadratic energy function. However, our analysis framework can be extended to more complicated cases and the insights can be expected to apply beyond our current setting.

**Related theoretical works.** Our analysis borrows proof techniques for analyzing algorithm properties from the optimization literature [18, 20] and for bounding Rademacher complexity from the statistical learning literature [13, 16, 17, 21, 22], but our focus and results are new. More precisely, the 'leave-

one-out' stability of optimization algorithms have been used to derive generalization bounds [23, 24, 25, 18, 26, 27]. However, all existing analyses are in the context where the optimization algorithms are used to train and select the model, while our analysis is based on a fundamentally different viewpoint where the algorithm itself is unrolled and integrated as a layer in the deep model. Also, existing works on the generalization of deep learning mainly focus on generic neural architectures such as feed-forward neural networks, RNN, GNN, etc [13, 14, 15]. The complexity of models based on highly structured iterative algorithms and the relation to algorithm properties have not been investigated. Furthermore, we are not aware of any previous use of local Rademacher complexity analysis for deep learning models.

## 2 Setting: Optimization Algorithms as Reasoning Modules

In many applications, reasoning can be accomplished by solving an optimization problem defined by a neural perceptual module. For instance, a visual SUDOKU puzzle can be solved using a neural module to perceive the digits followed by a quadratic optimization module to maximize a logic satisfiability objective [1]. The RNA folding problem can be tackled by a neural energy model to capture pairwise relations between RNA bases and a constrained optimization module to minimize the energy, with additional pairing constraints, to obtain a folding [2]. In a broader context, MAML [28, 29] also has a neural module for joint initialization and a reasoning module that performs optimization steps for task-specific adaptation. Other examples include [6, 30, 31, 32, 33, 34, 35, 36, 37, 38, 39, 40, 41, 42, 43]. More specifically, perception and reasoning can be jointly formulated in the form

$$\boldsymbol{y}(\boldsymbol{x}) = \arg\min_{\boldsymbol{y} \in \mathcal{Y}} E_\theta(\boldsymbol{x}, \boldsymbol{y}), \tag{1}$$

where $\boldsymbol{x}$ is an input, and $E_\theta(\boldsymbol{x}, \boldsymbol{y})$ is a neural energy function with parameters $\theta$, which specifies the type of information needed for performing reasoning, and together with constraints $\mathcal{Y}$ on the output $\boldsymbol{y}$, specifies the style of reasoning. Very often, the optimizer can be approximated by iterative algorithms, so the mapping in Eq. 1 can be approximated by the following end-to-end hybrid model

$$f_{\phi,\theta}(\boldsymbol{x}) := \mathtt{Alg}_\phi^k\left(E_\theta(\boldsymbol{x}, \cdot)\right) \ : \ \mathcal{X} \mapsto \mathcal{Y}. \tag{2}$$

$\mathtt{Alg}_\phi^k$ is the reasoning module with parameters $\phi$. Given a neural energy, it performs $k$-step iterative updates to produce the output (Fig 1). When $k$ is finite, $\mathtt{Alg}_\phi^k$ corresponds to approximate reasoning. As an initial attempt to analyze deep architectures with reasoning layers, we will restrict our analysis to a simple case where $E_\theta(\boldsymbol{x}, \boldsymbol{y})$ is quadratic in $\boldsymbol{y}$. A reason is that the analysis of advanced algorithms such as Nesterov accelerated gradients will become very complex for general cases. Similar problems occur in [18] which also restricts the proof to quadratic objectives. Specifically:

**Problem setting:** Consider a hybrid architecture where the neural module is an energy function of the form $E_\theta((\boldsymbol{x}, \boldsymbol{b}), \boldsymbol{y}) = \frac{1}{2}\boldsymbol{y}^\top Q_\theta(\boldsymbol{x})\boldsymbol{y} + \boldsymbol{b}^\top \boldsymbol{y}$, with $Q_\theta$ a neural network that maps $\boldsymbol{x}$ to a matrix. Each energy can be uniquely represented by $(Q_\theta(\boldsymbol{x}), \boldsymbol{b})$, so we can write the overall architecture as

$$f_{\phi,\theta}(\boldsymbol{x}, \boldsymbol{b}) := \mathtt{Alg}_\phi^k(Q_\theta(\boldsymbol{x}), \boldsymbol{b}). \tag{3}$$

Assume we are given a set of $n$ i.i.d. samples $S_n = \{((\boldsymbol{x}_1, \boldsymbol{b}_1), \boldsymbol{y}_1^*), \cdots, ((\boldsymbol{x}_n, \boldsymbol{b}_n), \boldsymbol{y}_n^*)\}$, where the labels $\boldsymbol{y}^*$ are given by the *exact minimizer* $\mathtt{Opt}$ of the corresponding $Q^*$, i.e.,

$$\boldsymbol{y}^* = \mathtt{Opt}(Q^*(\boldsymbol{x}), \boldsymbol{b}). \tag{4}$$

Then the learning problem is to find the best model $f_{\phi,\theta}$ from the space $\mathcal{F} := \{f_{\phi,\theta} : (\phi, \theta) \in \Phi \times \Theta\}$ by minimizing the empirical loss function

$$\min_{f_{\phi,\theta} \in \mathcal{F}} \ \frac{1}{n} \sum_{i=1}^{n} \ell_{\phi,\theta}(\boldsymbol{x}_i, \boldsymbol{b}_i), \tag{5}$$

where $\ell_{\phi,\theta}(\boldsymbol{x}, \boldsymbol{b}) := \|\mathtt{Alg}_\phi^k\left(Q_\theta(\boldsymbol{x}), \boldsymbol{b}\right) - \mathtt{Opt}(Q^*(\boldsymbol{x}), \boldsymbol{b})\|_2$. Furthermore, we assume:

• We have $\mathcal{Y} = \mathbb{R}^d$, and both $Q_\theta$ and $Q^*$ map $\mathcal{X}$ to $\mathcal{S}_{\mu,L}^{d \times d}$, the space of symmetric positive definite (SPD) matrices with $\mu, L > 0$ as its smallest and largest singular values, respectively. Thus the induced energy function $E_\theta$ will be $\mu$-strongly convex and $L$-smooth, and the output of $\mathtt{Opt}$ is unique.
• The input $(\boldsymbol{x}, \boldsymbol{b})$ is a pair of random variables where $\boldsymbol{x} \in \mathcal{X} \subseteq \mathbb{R}^m$ and $\boldsymbol{b} \in \mathcal{B} \subseteq \mathbb{R}^d$. Assume $\boldsymbol{b}$ satisfies $\mathbb{E}[\boldsymbol{b}\boldsymbol{b}^\top] = \sigma_b^2 I$. Assume $\boldsymbol{x}$ and $\boldsymbol{b}$ are independent, and their joint distribution follows a probability measure $P$. Assume samples in $S_n$ are drawn i.i.d. from $P$.

- Assume $\mathcal{B}$ is bounded, and let $M = \sup_{(Q,\boldsymbol{b}) \in \mathcal{S}_{\mu,L}^{d \times d} \times \mathcal{B}} \|\texttt{Opt}(Q, \boldsymbol{b})\|_2$.

Though this setting does not encompass the full complexity of hybrid deep architectures, it already reveals interesting connections between algorithm properties of the reasoning module and the learning behaviors of hybrid architectures.

## 3 Properties of Algorithms

In this section, we formally define the algorithm properties of the reasoning module $\texttt{Alg}_\phi^k$, under the problem setting presented in Sec 2. After that, we compare the corresponding properties of gradient descent, $\texttt{GD}_\phi^k$, and Nesterov's accelerated gradients, $\texttt{NAG}_\phi^k$, as concrete examples.

**(I)** The **convergence rate** of an algorithm expresses how fast the optimization error decreases as $k$ grows. Formally, we say $\texttt{Alg}_\phi^k$ has a convergence rate $Cvg(k, \phi)$ if for any $Q \in \mathcal{S}_{\mu,L}^{d \times d}, \boldsymbol{b} \in \mathcal{B}$,

$$\|\texttt{Alg}_\phi^k(Q, \boldsymbol{b}) - \texttt{Opt}(Q, \boldsymbol{b})\|_2 \leq Cvg(k, \phi)\|\texttt{Alg}_\phi^0(Q, \boldsymbol{b}) - \texttt{Opt}(Q, \boldsymbol{b})\|_2. \tag{6}$$

**(II) Stability** of an algorithm characterizes its robustness to small *perturbations in the optimization objective*, which corresponds to the perturbation of $Q$ and $\boldsymbol{b}$ in the quadratic case. For the purpose of this paper, we say an algorithm $\texttt{Alg}_\phi^k$ is $Stab(k, \phi)$-stable if for any $Q, Q' \in \mathcal{S}_{\mu,L}^{d \times d}$ and $\boldsymbol{b}, \boldsymbol{b}' \in \mathcal{B}$,

$$\|\texttt{Alg}_\phi^k(Q, \boldsymbol{b}) - \texttt{Alg}_\phi^k(Q', \boldsymbol{b}')\|_2 \leq Stab(k, \phi)\|Q - Q'\|_2 + Stab(k, \phi)\|\boldsymbol{b} - \boldsymbol{b}'\|_2, \tag{7}$$

where $\|Q - Q'\|_2$ is the spectral norm of the matrix $Q - Q'$.

**(III) Sensitivity** characterizes the robustness to small *perturbations in the algorithm parameters* $\phi$. We say the sensitivity of $\texttt{Alg}_\phi^k$ is $Sens(k)$ if it holds for all $Q \in \mathcal{S}_{\mu,L}^{d \times d}, \boldsymbol{b} \in \mathcal{B}$, and $\phi, \phi' \in \Phi$ that

$$\|\texttt{Alg}_\phi^k(Q, \boldsymbol{b}) - \texttt{Alg}_{\phi'}^k(Q, \boldsymbol{b})\|_2 \leq Sens(k)\|\phi - \phi'\|_2. \tag{8}$$

This concept is referred in the deep learning community to "parameter perturbation error" or "sharpness" [44, 45, 46]. It has been used for deriving generalization bounds of neural networks, both in the Rademacher complexity framework [13] and PAC-Bayes framework [47].

**(IV)** The **stable region** is the range $\Phi$ of the parameters $\phi$ where the algorithm output will remain bounded as $k$ to infinity, i.e., numerically stable. Only when the algorithms operate in the stable region, the corresponding $Cvg(k, \phi)$, $Stab(k, \phi)$ and $Sens(k)$ will remain finite for all $k$. It is usually very difficult to identity the exact stable region, but a sufficient range can be provided.

**GD and NAG.** Now we will compare the above four algorithm properties for gradient descent and Nesterov's accelerated gradient method, both of which can be used to solve the quadratic optimization in our problem setting. First, the algorithm update steps are summarized bellow:

$$\texttt{GD}_\phi : \boldsymbol{y}_{k+1} \leftarrow \boldsymbol{y}_k - \phi(Q\boldsymbol{y}_k + \boldsymbol{b}) \qquad \texttt{NAG}_\phi : \begin{cases} \boldsymbol{y}_{k+1} \leftarrow \boldsymbol{z}_k - \phi(Q\boldsymbol{z}_k + \boldsymbol{b}) \\ \boldsymbol{z}_{k+1} \leftarrow \boldsymbol{y}_{k+1} + \frac{1 - \sqrt{\mu\phi}}{1 + \sqrt{\mu\phi}}(\boldsymbol{y}_{k+1} - \boldsymbol{y}_k) \end{cases} \tag{9}$$

where the hyperparameter $\phi$ corresponds to the step size. The initializations $\boldsymbol{y}_0, \boldsymbol{z}_0$ are set to zero vectors throughout this paper. Denote the results of $k$-step update, $\boldsymbol{y}_k$, of GD and NAG by $\texttt{GD}_\phi^k(Q, \boldsymbol{b})$ and $\texttt{NAG}_\phi^k(Q, \boldsymbol{b})$, respectively. Then their algorithm properties are summarized in Table 1.

Table 1: Comparison of algorithm properties between GD and NAG. For simplicity, only the order in $k$ is presented. Complete statements with detailed coefficients and proofs are given in Appendix A.

| Alg | $Cvg(k, \phi)$ | $Stab(k, \phi)$ | $Sens(k)$ | Stable region $\Phi$ |
|---|---|---|---|---|
| $\texttt{GD}_\phi^k$ | $\mathcal{O}\left((1 - \phi\mu)^k\right)$ | $\mathcal{O}\left(1 - (1 - \phi\mu)^k\right)$ | $\mathcal{O}\left(k(1 - c_0\mu)^{k-1}\right)$ | $[c_0, \frac{2}{\mu+L}]$ |
| $\texttt{NAG}_\phi^k$ | $\mathcal{O}\left(k(1 - \sqrt{\phi\mu})^k\right)$ | $\mathcal{O}\left(1 - (1 - \sqrt{\phi\mu})^k\right)$ | $\mathcal{O}\left(k^3(1 - \sqrt{c_0\mu})^k\right)$ | $[c_0, \frac{4}{\mu+3L}]$ |

Table 1 shows: *(i) Convergence*: NAG converges faster than GD, especially when $\mu$ is very small, which is a well-known result. *(ii) Stability*: However, as $k$ grows, NAG is less stable than GD for a fixed $k$, in contrast to their convergence behaviors. This is pointed out in [18], which proves that a faster converging algorithm has to be less stable. *(iii) Sensitivity*: The sensitivity behaves similar to

the convergence, where NAG is less sensitive to step-size perturbation than GD. Also, the sensitivity of both algorithms gets smaller as $k$ grows larger. *(iv): Stable region*: Since $\mu < L$, the stable region of GD is larger than that of NAG. It means a larger step size is allowable for GD that will not lead to exploding outputs even if $k$ is large. Note that all the other algorithm properties are based on the assumption that $\phi$ is in the stable region $\Phi$. Furthermore, as $k$ goes to infinity, the space $\{\texttt{Alg}_\phi^k : \phi \in \Phi\}$ will finally shrink to a single function, which is the exact minimizer $\{\texttt{Opt}\}$.

Our purpose of comparing the algorithm properties of GD and NAG is to show in a later section their difference as a reasoning layer in deep architectures. However, some results in Table 1 are new by themselves, which may be of independent interest. For instance, we are not aware of other analysis of the sensitivity of GD and NAG to their step-size perturbation. Besides, for the stability results, we provide a proof with a weaker assumption where $\phi$ can be larger than $1/L$, which is not allowed in [18]. This is necessary since in practice the learned step size $\phi$ is usually larger than $1/L$.

## 4  Approximation Ability

How will the algorithm properties affect the approximation ability of deep architecture with reasoning layers? Given a model space $\mathcal{F} := \{\texttt{Alg}_\phi^k(Q_\theta(\cdot), \cdot) : \phi \in \Phi, \theta \in \Theta\}$, we are interested in its approximation ability to functions of the form $\texttt{Opt}\,(Q^*(\boldsymbol{x}), \boldsymbol{b})$. More specifically, we define the loss

$$\ell_{\phi,\theta}(\boldsymbol{x}, \boldsymbol{b}) := \|\texttt{Alg}_\phi^k(Q_\theta(\boldsymbol{x}), \boldsymbol{b}) - \texttt{Opt}(Q^*(\boldsymbol{x}), \boldsymbol{b})\|_2, \tag{10}$$

and measure the approximation ability by $\inf_{\phi \in \Phi, \theta \in \Theta} \sup_{Q^* \in \mathcal{Q}^*} P\ell_{\phi,\theta}$, where $\mathcal{Q}^* := \{\mathcal{X} \mapsto \mathcal{S}_{\mu,L}^{d \times d}\}$ and $P\ell_{\phi,\theta} = \mathbb{E}_{\boldsymbol{x}, \boldsymbol{b}}[\ell_{\phi,\theta}(\boldsymbol{x}, \boldsymbol{b})]$. Intuitively, using a faster converging algorithm, the model $\texttt{Alg}_\phi^k$ could represent the reasoning-task structure, $\texttt{Opt}$, better and improve the overall approximation ability. Indeed we can prove the following lemma confirming this intuition.

**Lemma 4.1.  (Faster Convergence $\Rightarrow$ Better Approximation Ability).** *Assume the problem setting in Sec 2. The approximation ability can be bounded by two terms:*

$$\inf_{\phi \in \Phi, \theta \in \Theta} \sup_{Q^* \in \mathcal{Q}^*} P\ell_{\phi,\theta} \le \sigma_b \mu^{-2} \underbrace{\inf_{\theta \in \Theta} \sup_{Q^* \in \mathcal{Q}^*} P\|Q_\theta - Q^*\|_F}_{\textit{approximation ability of the neural module}} + M \underbrace{\inf_{\phi \in \Phi} Cvg(k, \phi)}_{\textit{best convergence}}. \tag{11}$$

With Lemma 4.1, we conclude that: A faster converging algorithm can define a model with better approximation ability. For example, for a fixed $k$ and $Q_\theta$, NAG converges faster than GD, so $\texttt{NAG}_\phi^k$ can approximate $\texttt{Opt}$ more accurately than $\texttt{GD}_\phi^k$, which is experimentally validated in Sec 7.

Similarly, we can also reverse the reasoning, and ask the question that, given two hybrid architectures with the same approximation error, which architecture has a smaller error in representing the energy function $Q^*$? We show that this error is also intimately related to the convergence of the algorithm.

**Lemma 4.2.  (Faster Convergence $\Rightarrow$ Better Representation of $Q^*$).** *Assume the problem setting in Sec 2. Then $\forall \phi \in \Phi, \theta \in \Theta, Q^* \in \mathcal{Q}^*$ it holds true that*

$$P\|Q_\theta - Q^*\|_F^2 \le \sigma_b^{-2} L^4 (\sqrt{P\ell_{\phi,\theta}^2} + M \cdot Cvg(k, \phi))^2. \tag{12}$$

Lemma 4.2 highlights the benefit of using an algorithmic layer that aligns with the reasoning-task structure. Here the task structure is represented by $\texttt{Opt}$, the minimizer, and convergence measures how well $\texttt{Alg}_\phi^k$ is aligned with $\texttt{Opt}$. Lemma 4.2 essentially indicates that *if the structure of a reasoning module can better align with the task structure, then it can better constrain the search space of the underlying neural module $Q_\theta$, making it easier to learn, and further lead to better sample complexity,* which we will explain more in the next section.

As a concrete example for Lemma 4.2, if $\texttt{GD}_\phi^k(Q_\theta, \cdot)$ and $\texttt{NAG}_\phi^k(Q_\theta, \cdot)$ achieve the **same** accuracy for approximating $\texttt{Opt}\,(Q^*, \cdot)$, then the neural module $Q_\theta$ in $\texttt{NAG}_\phi^k(Q_\theta, \cdot)$ will have a **better** accuracy for approximating $Q^*$ than $Q_\theta$ in $\texttt{GD}_\phi^k(Q_\theta, \cdot)$. In other words, a faster converging algorithm imposes more constraints on the energy function $Q_\theta$, making it approach $Q^*$ faster.

# 5 Generalization Ability

How will algorithm properties affect the generalization ability of deep architectures with reasoning layers? We theoretically showed that the generalization bound is determined by both the algorithm properties and the complexity of the neural module. Moreover, it induces interesting implications - when the neural module is over- or under- parameterized, the generalization bound is dominated by algorithm stability; but when the neural module has an about-right parameterization, the bound is dominated by the product of algorithm stability and convergence.

More specifically, we will analyze *generalization gap* between the expected loss and empirical loss,

$$P\ell_{\phi,\theta} = \mathbb{E}_{\boldsymbol{x},\boldsymbol{b}}\ell_{\phi,\theta}(\boldsymbol{x},\boldsymbol{b}) \text{ and } P_n\ell_{\phi,\theta} = \frac{1}{n}\sum_{i=1}^n \ell_{\phi,\theta}(\boldsymbol{x}_i,\boldsymbol{b}_i), \text{ respectively,} \quad (13)$$

where $P_n$ is the empirical probability measure induced by the samples $S_n$. Let $\ell_{\mathcal{F}} := \{\ell_{\phi,\theta} : \phi \in \Phi, \theta \in \Theta\}$ be the function space of losses of the models. The generalization gap, $P\ell_{\phi,\theta} - P_n\ell_{\phi,\theta}$, can be bounded by the Rademacher complexity, $\mathbb{E}R_n\ell_{\mathcal{F}}$, which is defined as the expectation of the empirical Rademacher complexity, $R_n\ell_{\mathcal{F}} := \mathbb{E}_{\boldsymbol{\sigma}} \sup_{\phi\in\Phi,\theta\in\Theta} \frac{1}{n}\sum_{i=1}^n \sigma_i\ell_{\phi,\theta}(\boldsymbol{x}_i,\boldsymbol{b}_i)$, where $\{\sigma_i\}_{i=1}^n$ are $n$ independent Rademacher random variables uniformly distributed over $\{\pm 1\}$. Generalization bounds derived from Rademacher complexity have been studied in many works [48, 49, 50, 51].

However, deriving the Rademacher complexity of $\ell_{\mathcal{F}}$ is highly nontrivial in our case, and we are not aware of prior bounds for deep learning models with reasoning layers. Aiming at bridging the relation between algorithm properties and generalization ability that can explain experimental observations, we find that standard Rademacher complexity analysis is insufficient. The shortcoming of the standard Rademacher complexity is that it provides *global* estimates of the complexity of the model space, which ignores the fact that the training process will likely pick models with small errors. Taking this factor into account, we resort to more refined analysis using *local* Rademacher complexity [13, 16, 17]. Remarkably, we found that the bounds derived via *global* and *local* Rademacher complexity will lead to *different* conclusions about the effects of algorithm layers. That is, an algorithm that converges faster could lead to a model space that has a larger global Rademacher complexity but a smaller local Rademacher complexity. Also, the *global* Rademacher complexity is dominated by algorithmic stability. However, in the *local* counterpart, there is a trade-off term between stability and convergence, which aligns better with the experimental observations.

**Main Result.** More specifically, the local Rademacher complexity of $\ell_{\mathcal{F}}$ at level $r$ is defined as

$$\mathbb{E}R_n\ell_{\mathcal{F}}^{loc}(r) \text{ where } \ell_{\mathcal{F}}^{loc}(r) := \{\ell_{\phi,\theta} : \phi \in \Phi, \theta \in \Theta, P\ell_{\phi,\theta}^2 \leq r\}. \quad (14)$$

This notion is less general than the one defined in [16, 17] but is sufficient for our purpose. Here we also define a loss function space $\ell_{\mathcal{Q}} := \{\|Q_\theta - Q^*\|_F : \theta \in \Theta\}$ for the neural module $Q_\theta$, and introduce its local Rademacher complexity $\mathbb{E}R_n\ell_{\mathcal{Q}}^{loc}(r_q)$, where $\ell_{\mathcal{Q}}^{loc}(r_q) = \{\|Q_\theta - Q^*\|_F \in \ell_{\mathcal{Q}} : P\|Q_\theta - Q^*\|_F^2 \leq r_q\}$. With these definitions, we can show that the local Rademacher complexity of the hybrid architecture is explicitly related to all considered algorithm properties, namely convergence, stability and sensitivity, and there is an intricate trade-off.

**Theorem 5.1.** *Assume the problem setting in Sec 2. Then we have for any $t > 0$ that*

$$\mathbb{E}R_n\ell_{\mathcal{F}}^{loc}(r) \leq \sqrt{2}dn^{-\frac{1}{2}}Stab(k)\left(\sqrt{(Cvg(k)M + \sqrt{r})^2 C_1(n) + C_2(n,t)} + C_3(n,t) + 4\right) \quad (15)$$

$$+ Sens(k)B_\Phi, \quad (16)$$

*where $Stab(k) = \sup_\phi Stab(k,\phi)$ and $Cvg(k) = \sup_\phi Cvg(k,\phi)$ are worst-case stability and convergence, $B_\Phi = \frac{1}{2}\sup_{\phi,\phi'\in\Phi}\|\phi - \phi'\|_2$, $C_1(n) = \mathcal{O}(\log N(n))$, $C_3(n,t) = \mathcal{O}(\frac{\log N(n)}{\sqrt{n}} + \frac{\sqrt{\log N(n)}}{e^t})$, $C_2(n,t) = \mathcal{O}(\frac{t\log N(n)}{n} + (C_3(n,t) + 1)\frac{\log N(n)}{\sqrt{n}})$, and $N(n) = \mathcal{N}(\frac{1}{\sqrt{n}}, \ell_{\mathcal{Q}}, L_\infty)$ is the covering number of $\ell_{\mathcal{Q}}$ with radius $\frac{1}{\sqrt{n}}$ and $L_\infty$ norm.*

*Proof Sketch.* We will explain the key steps here, and the full proof details are deferred to Appendix C. The essence of the proof is to find the relation between $R_n\ell_{\mathcal{F}}^{loc}(r)$ and $R_n\ell_{\mathcal{Q}}^{loc}(r_q)$, and also the relation between the local level $r$ and $r_q$. Then the analysis of the local Rademachar complexity of the end-to-end model $\text{Alg}_\phi^k(Q_\theta, \cdot)$ can be reduced to that of the neural module $Q_\theta$.

More specifically, we first show that the loss $\ell_{\phi,\theta}$ is $Stab(k)$-Lipschitz in $Q_\theta$ and $Sens(k)$-Lipschitz in $\phi$. By the triangle inequality and algorithm properties, we can bound the sensitivity of the loss by

$$|\ell_{\phi,\theta}(\boldsymbol{x}) - \ell_{\phi',\theta'}(\boldsymbol{x})| \leq Stab(k)\|Q_\theta(\boldsymbol{x}) - Q_{\theta'}(\boldsymbol{x})\|_2 + Sens(k)\|\phi - \phi'\|_2. \quad (17)$$

Second, by leveraging vector-contraction inequality for Rademacher complexity of vector-valued hypothesis [21, 22] and our previous observations in Lemma 4.2, we can turn the sensitivity bound on the loss function in Eq. 17 to a local Rademacher complexity bound

$$R_n\ell_{\mathcal{F}}^{loc}(r) \leq \sqrt{2}d\,Stab(k)R_n\ell_{\mathcal{Q}}^{loc}(r_q) + Sens(k)B_\Phi \text{ with } r_q = \sigma_b^{-2}L^4(\sqrt{r} + MCvg(k))^2. \quad (18)$$

Therefore, bounding the local Rademacher complexity of $\ell_{\mathcal{F}}^{loc}$ at level $r$ resorts to bounding that of $\ell_{\mathcal{Q}}^{loc}$ at level $r_q$. This inequality has already revealed the role of stability, convergence, and sensitivity in bounding local Rademacher complexity, and is the key step in the proof.

Third, based on an extension of Talagrand's inequality for empirical processes [52, 16], we can bound the empirical error $P_n\|Q_\theta - Q^*\|_F^2$ using $r_q$ and some other terms with high probability. Then $R_n\ell_{\mathcal{Q}}^{loc}(r_q)$ can be bounded using the covering number of $\ell_{\mathcal{Q}}$ via the classical Dudley entropy integral [53], where the upper integration bound is given by the upper bound of $P_n\|Q_\theta - Q^*\|_F^2$. $\qquad\square$

**Trade-offs between convergence, stability and sensitivity.** Generally speaking, the convergence rate $Cvg(k)$ and sensitivity $Sens(k)$ have similar behavior, but $Stab(k)$ behaves opposite to them; see illustrations in Fig 2. Therefore, the way these three quantities interact in Theorem 5.1 suggests that in different regimes one may see different generalization behavior. More specially, depending on the parameterization of $Q_\theta$, the coefficients $C_1$, $C_2$, and $C_3$ in Eq. 15 may have different scale, making the local Rademacher complexity bound dominated by different algorithm properties. Since the coefficients $C_i$ are monotonely increasing in the covering number of $\ell_{\mathcal{Q}}$, we expect that:

**(i)** When $Q_\theta$ is **over-parameterized**, the covering number of $\ell_{\mathcal{Q}}$ becomes large, as do the three coefficients. Large $C_i$ will reduce the effect of $Cvg(k)$ and make Eq. 15 dominated by $Stab(k)$;

**(ii)** When $Q_\theta$ is **under-parameterized**, the three coefficients get small, but they still reduce the effect of $Cvg(k)$ given the constant 4 in Eq. 15, again making it dominated by $Stab(k)$;

**(iii)** When the parametrization of $Q_\theta$ is **about-right**, we can expect $Cvg(k)$ to play a critical role in Eq. 15, which will then behave similar to the product $Stab(k)Cvg(k)$, as illustrated schematically in Fig 2. We experimentally validate these implications in Sec 7.

**Trade-off of the depth.** Combining the above implications with the approximation ability analysis in Sec 4, we can see that in the above-mentioned cases **(i)** and **(ii)**, deeper algorithm layers will lead to better approximation accuracy but worse generalization. Only in the ideal case

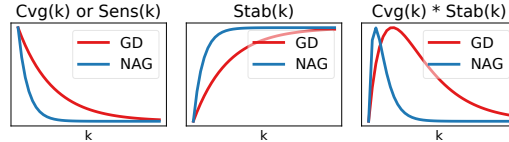

Figure 2: Overall trend of algorithm properties.

**(iii)**, a deeper reasoning module can induce both better representation and generalization abilities. This result provides practical guidelines for some recently proposed infinite-depth models [54, 55].

**Standard Rademacher complexity analysis.** If we consider the standard Rademacher complexity and directly bound it by the covering number of $\ell_{\mathcal{F}}$ via Dudley's entropy integral in the way some existing generalization bounds of deep learning are derived [13, 14, 15], we will get the following upper bound for the covering number, where $Cvg(k)$ does not play a role:

$$\mathcal{N}(\epsilon, \ell_{\mathcal{F}}, L_2(P_n)) \leq \mathcal{N}(\epsilon/(2Stab(k)), \mathcal{Q}, L_2(P_n)) \cdot \mathcal{N}(\epsilon/(2Sens(k)), \Phi, \|\cdot\|_2). \quad (19)$$

Since $\Phi$ only contains the hyperparameters in the algorithm and $\mathcal{Q} := \{Q_\theta, \theta \in \Theta\}$ is often highly expressive, typically stability will dominate this bound. Or, consider the case when algorithm layers are fixed so $\Phi$ only contains one element. Then this covering number is determined by stability, which infers that $\mathtt{NAG}_1^k(Q_\theta, \cdot)$ has a **larger** Rademacher complexity than $\mathtt{GD}_1^k(Q_\theta, \cdot)$ since it is less stable. However, in the local Rademacher complexity bound in Theorem 5.1, even if $Sens(k)$ in Eq. 16 is ignored, there is still a trade-off between convergence and stability which implies $\mathtt{NAG}_1^k(Q_\theta, \cdot)$ can have a **smaller** local Rademacher complexity than $\mathtt{GD}_1^k(Q_\theta, \cdot)$, leading to a different conclusion. Our experiments show the local Rademacher complexity bound is better for explaining the actual observations.

# 6  Pros and Cons of RNN as a Reasoning Layer

It has been shown that RNN (or GNN) can represent reasoning and iterative algorithms over structures [19, 15]. Can our analysis framework also be used to understand RNN (or GNN)? How will its behavior compare with more interpretable algorithm layers such as $\texttt{GD}_\phi^k$ and $\texttt{NAG}_\phi^k$? In the case of RNN, the algorithm update steps in each iteration are given by an RNN cell

$$\boldsymbol{y}_{k+1} \leftarrow \texttt{RNNcell}\left(Q, \boldsymbol{b}, \boldsymbol{y}_k\right) := V\sigma\left(W^L\sigma\left(W^{L-1}\cdots W^2\sigma\left(W_1^1\boldsymbol{y}_t + W_2^1\boldsymbol{g}_t\right)\right)\right). \quad (20)$$

where the activation function $\sigma =$ ReLU takes $\boldsymbol{y}_k$ and the gradient $\boldsymbol{g}_t = Q\boldsymbol{y}_t + \boldsymbol{b}$ as inputs. Then a recurrent neural network $\texttt{RNN}_\phi^k$ having $k$ unrolled RNN cells can be viewed as a neural algorithm. The algorithm properties of $\texttt{RNN}_\phi^k$ are summarized in Table 2. Assume $\phi = \{V, W_1^1, W_2^1, W^{2:L}\}$ is in a stable region with $c_\phi := \sup_Q \|V\|_2 \|W_1^1 + W_2^1 Q\|_2 \prod_{l=2}^L \|W^l\|_2 < 1$, so that the operations in $\texttt{RNNcell}$ are strictly contractive, i.e., $\|\boldsymbol{y}_{k+1} - \boldsymbol{y}_k\|_2 < \|\boldsymbol{y}_k - \boldsymbol{y}_{k-1}\|_2$. In this case, the stability and sensitivity of $\texttt{RNN}_\phi^k$ are guaranteed to be bounded.

However, the fundamental disadvantage of RNN is its lack of worst-case guarantee for convergence. In general the outputs of $\texttt{RNN}_\phi^k$ may not converge to the minimizer $\texttt{Opt}$, meaning that its worst-case convergence rate can be much larger than 1. This will lead to worse generalization bound according to our theory compared to $\texttt{GD}_\phi^k$ and $\texttt{NAG}_\phi^k$.

Table 2: Properties of $\texttt{RNN}_\phi^k$. (Details are given in Appendix D.)

| Stable region $\Phi$ | $c_\phi < 1$ |
|---|---|
| $Stab(k, \phi)$ | $\mathcal{O}(1 - c_\phi^k)$ |
| $Sens(k)$ | $\mathcal{O}(1 - (\inf_\phi c_\phi)^k)$ |
| $\min_\phi Cvg(k, \phi)$ | $\mathcal{O}(\rho^k)$ with $\rho < 1$ |

The advantage of RNN is its expressiveness, especially given the universal approximation ability of MLP in the RNNcell. One can show that $\texttt{RNN}_\phi^k$ can express $\texttt{GD}_\phi^k$ or $\texttt{NAG}_\phi^k$ with suitable choices of $\phi$. Therefore, its best-case convergence can be as small as $\mathcal{O}(\rho^k)$ for some $\rho < 1$. When the needed types of reasoning is unknown or beyond what existing algorithms are capable of, RNN has the potential to learn new reasoning types given sufficient data.

# 7  Experimental Validation

Our experiments aim to validate our theoretical prediction with computational simulations, rather than obtaining state-of-the-art results. We hope the theory together with these experiments can lead to practical guidelines for designing deep architectures with reasoning layers. We conduct two sets of experiments, where the first set of experiments strictly follows the problem setting described in Sec 2 and the second is conducted on BSD500 dataset [56] to demonstrate the possibility of generalizing the theorem to more realistic applications. Implementations in Python are released[1].

## 7.1  Synthetic Experiments

The experiments follow the problem setting in Sec 2. We sample 10000 pairs of $(\boldsymbol{x}, \boldsymbol{b})$ uniformly as overall dataset. During training, $n$ samples are randomly drawn from these 10000 data points as the training set. Each $Q^*(\boldsymbol{x})$ is produced by a rotation matrix and a vector of eigenvalues parameterized by a randomly fixed 2-layer dense neural network with hidden dimension 3. Then the labels are generated according to $\boldsymbol{y} = \texttt{Opt}(Q^*(\boldsymbol{x}), \boldsymbol{b})$. We train the model $\texttt{Alg}_\phi^k(Q_\theta, \cdot)$ on $S_n$ using the loss in Eq. 10. Here, $Q_\theta$ has the same overall architecture as $Q^*$ but the hidden dimension could vary. Note that in all figures, each $k$ corresponds to an **independently trained model** with $k$ iterations in the algorithm layer, instead of the sequential outputs of a single model. Each model is trained by ADAM and SGD with learning rate grid-searched from [1e-2,5e-3,1e-3,5e-4,1e-4], and only the best result is reported. Furthermore, error bars are produced by 20 independent instantiations of the experiments. See Appendix E for more details.

**Approximation ability.** To validate Lemma 4.1, we compare $\texttt{GD}_\phi^k(Q_\theta, \cdot)$ and $\texttt{NAG}_\phi^k(Q_\theta, \cdot)$ in terms of approximation accuracy. For various hidden sizes of $Q_\theta$, the results are similar, so we report one representative case in Fig 3. The approximation accuracy aligns with the convergence of the algorithms, showing that faster converging algorithm can induce better approximation ability.

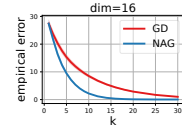

Figure 3

**Faster convergence⇒better** $Q_\theta$**.** We report the error of the neural module $Q_\theta$ in Fig 4. Note that $\texttt{Alg}_\phi^k(Q_\theta, \cdot)$ is trained end-to-end, without supervision on $Q_\theta$. In Fig 4, the error of $Q_\theta$ decreases as

$k$ grows, in a rate similar to algorithm convergence. This validates the implication of Lemma 4.2 that, when $\text{Alg}_\phi^k$ is closer to $\text{Opt}$, it can help the underlying neural module $Q_\theta$ to get closer to $Q^*$.

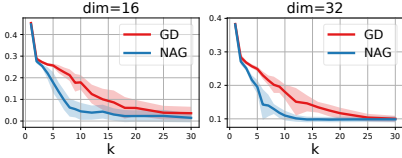
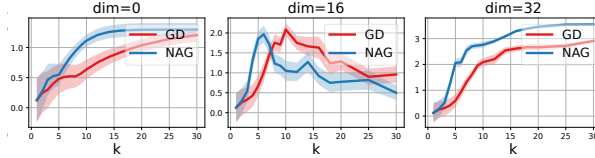

Figure 4: $P\|Q_\theta - Q^*\|_F^2$          Figure 5: Generalization gap

**Generalization gap.** In Fig 5, we report the generalization gaps, with hidden sizes of $Q_\theta$ being 0, 16, and 32, which corresponds to the three cases **(ii)**, **(iii)**, and **(i)** discussed under Theorem 5.1, respectively. Comparing Fig 5 to Fig 2, we can see that the experimental results match very well with the theoretical implications.

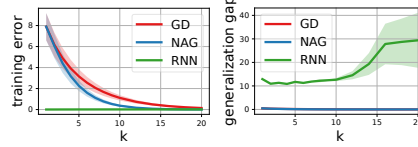

Figure 6: Algorithm layers vs RNN.

**RNN**. As discussed in Sec 6, RNN can be viewed as neural algorithms. To have a cleaner comparison, we report their behaviors under the 'learning to optimize' senario where the objectives $(Q, \boldsymbol{b})$ are given. Fig 6 shows that RNN has a better representation power but worse generalization ability.

## 7.2 Experiments on Real Dataset

To show the real world applicability of our theoretical framework, we consider the **local adaptive image denoising** task. Details are given below.

**Dataset.** We split BSD500 (400 images) into a training set (100 images) and a test set (300 images). Gaussian noises are added to each pixel with noise levels depending on image local smoothness, making the noise levels on edges lower than non-edge regions. The task is to restore the original image from the noisy version $X \in [0,1]^{180 \times 180}$.

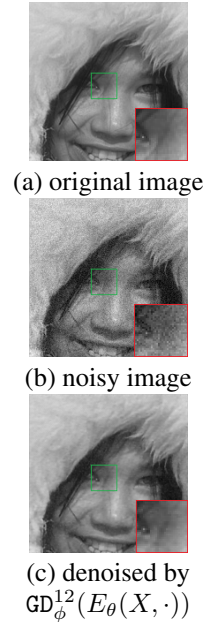

(a) original image

(b) noisy image

(c) denoised by $\text{GD}_\phi^{12}(E_\theta(X, \cdot))$

**Architecture.** In $\text{Alg}_\phi^k(E_\theta(X, \cdot))$, $\text{Alg}_\phi^k$ is a $k$-step unrolled minimization algorithm to the $\ell_2$-regularized reconstruction objective $E_\theta(X, Y) := \frac{1}{2}\|Y + g_\theta(X) - X\|_F^2 + \frac{1}{2}\sum_{i,j} |[f_\theta(X)]_{i,j} Y_{i,j}|^2$, and the residual $g_\theta(X)$ and position-wise regularization coefficient $f_\theta(X)$ are both DnCNN networks as in [57]. The optimization objective, $E_\theta(X, Y)$, is quadratic in $Y$.

**Generalization gap.** We instantiate the hybrid architecture into different models using GD and NAG algorithms with different unrolled steps $k$. Each model is trained with 3000 epochs, and the *generalization gaps* are reported in Fig. 7. The results also show good consistency with our theory, where stabler algorithm (GD) can generalize better given *over/under*-parameterized neural module, and for the *about-right* parameterization case, the generalization gap behaviors are similar to $Stab(k) * Cvg(k)$.

**Visualization.** To show that the learned hybrid model has a good performance in this real application, we include a visualization of the original, noisy, and denoised images.

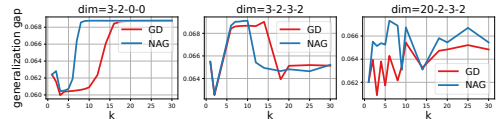

Figure 7: Generalization gap. **Each $k$ corresponds to a separately trained model**. *Left* (underparameterized): $f_\theta$ is a DnCNN with 3 channels and 2 hidden layers and $g_\theta = 0$. *Middle* (about-right): both $f_\theta$ and $g_\theta$ have 3 channels and 2 hidden layers. *Right (over-parameterized)*: $f_\theta$ has 20 channels.

## 8 Conclusion and Discussion

In this paper, we take an initial step towards the theoretical understanding of deep architectures with reasoning layers. Our theorem indicates intriguing relation between algorithm properties of the reasoning module and the approximation and generalization of the end-to-end model, which in turn provides practical guideline for designing reasoning layers. The current analysis is limited due to the simplified problem setting. However, assumptions we made are only for avoiding the non-uniqueness of the reasoning solution and the instability of the mapping from the reasoning solution to the neural module. The assumptions could be relaxed if we can involve other techniques to resolve these issues. These additional efforts could potentially generalize the results to more complex cases.

## Broader Impact

A common ethical concern of deep learning models is that they may not perform well on unseen examples, which could lead to the risk of producing biased content reflective of the training data. Our work, which learns an energy optimization model from the data, is not an exception. The approach we adopt to address this issue is to design hybrid deep architectures containing specialized reasoning modules. In the setting of quadratic energy functions, our theoretical analysis and numerical experiments show that hybrid deep models produce more reliable results than generic deep models on unseen data sets. More work is needed to determine the extent to which such hybrid model prevents biased outputs in more sophisticated tasks

## Acknowledgement

We would like to thank Professor Vladimir Koltchinskii for providing valuable suggestions and thank anonymous reviewers for providing constructive feedbacks. This work is supported in part by NSF grants CDS&E-1900017 D3SC, CCF-1836936 FMitF, IIS-1841351, CAREER IIS-1350983 to L.S.

## Footnotes

[1]https://github.com/xinshi-chen/Deep-Architecture-With-Reasoning-Layer

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
