[Supplementary Material]

# A   Proof of Algorithm Properties

In this section, we study several important properties of gradient descent algorithm (GD) and Nesterov's accelerated gradient algorithm (NAG), which have already been summarized in Table 1 of Section 3. To simplify the presentation, we shall focus on quadratic minimization problems as in Section 2 and estimate the sharp dependence on the iteration number $k$.

More precisely, in the subsequent analysis, we shall fix the constants $L \geq \mu > 0$ and assume the objective function is in the function class $\mathcal{Q}_{\mu,L}$, which contains all $\mu$-strongly convex and $L$-smooth quadratic functions on $\mathbb{R}^d$. Then, for any given $f \in \mathcal{Q}_{\mu,L}$, the eigenvalue decomposition enables us to represent the Hessian matrix of $f$, denoted by $Q$, as $Q = U\Lambda U^\top$, where $\Lambda$ is a diagonal matrix comprising of the eigenvalues $(\lambda_i)_{i=1}^d$ of $Q$ sorted in ascending order, i.e., $\mu \leq \lambda_1 \leq \ldots \leq \lambda_d \leq L$, and $U \in \mathbb{R}^{d \times d}$ is an orthogonal matrix whose columns constitute an orthonormal basis of corresponding eigenvectors of $Q$. Moreover, we shall denote by $\mathbb{I}_d$ the $d \times d$ identity matrix, and by $\|A\|_2$ the spectral norm of a given matrix $A \in \mathbb{R}^{d \times d}$.

We start with the GD algorithm. Let $f \in \mathcal{Q}_{\mu,L}$, $s \geq 0$ be the stepsize, and $x_0 \in \mathbb{R}^d$ be the initial guess. For each $k \in \mathbb{N} \cup \{0\}$, we denote by $x_{k+1}$ the $k+1$-th iterate generated by the following recursive formula (cf. the output $\boldsymbol{y}_{k+1}$ of $\mathtt{GD}_\phi$ in Section 3):

$$x_{k+1} = x_k - s\nabla f(x_k). \tag{21}$$

The following theorem establishes the convergence of Eq. 21 as $k$ tends to infinity, and the Lipschitz dependence of the iterates $(x_k^s)_{k \in \mathbb{N}}$ in terms of the stepsize $s$ (i.e., the sensitivity of GD). Similar results can be established for general $\mu$-strongly convex and $L$-smooth objective functions.

**Theorem A.1.** *Let $f \in \mathcal{Q}_{\mu,L}$ admit the minimiser $x^* \in \mathbb{R}^d$, $x_0 \in \mathbb{R}^d$ and for each $s \geq 0$ let $(x_k^s)_{k \in \mathbb{N} \cup \{0\}}$ be the iterates generated by Eq. 21 with stepsize $s$. Then we have for all $k \in \mathbb{N}$, $c_0 > 0$, $s, t \in [c_0, \frac{2}{\mu+L}]$ that*

$$\|x_k^s - x^*\|_2 \leq (1 - s\mu)^k \|x_0 - x^*\|_2, \quad \|x_k^t - x_k^s\|_2 \leq Lk(1 - c_0\mu)^{k-1}|t - s|\|x_0 - x^*\|_2. \tag{22}$$

*Proof.* Let $Q$ be the Hessian matrix of $f$ and $(\lambda_i)_{i=1}^d$ be the eigenvalues of $Q$. By using the fact that $\nabla f(x^*) = 0$ and Eq. 21, we can obtain for all $k \in \mathbb{N} \cup \{0\}$ and $s \geq 0$ that $x_k^s - x^* = (\mathbb{I}_d - sQ)(x_{k-1}^s - x^*) = (\mathbb{I}_d - sQ)^k(x_0 - x^*)$.

Since the spectral norm of a matrix is invariant under orthogonal transformations, we have for all $s \in [c_0, \frac{2}{\mu+L}]$ that

$$\begin{aligned}
\|\mathbb{I}_d - sQ\|_2 = \|\mathbb{I}_d - s\Lambda\|_2 &= \max_{i=1,\ldots,d}|1 - s\lambda_i| = \max(|1 - s\mu|, |1 - sL|) \\
&\leq 1 - s\mu.
\end{aligned} \tag{23}$$

Hence, for any given $k \in \mathbb{N} \cup \{0\}$, the inequality that $\|x_k^s - x^*\|_2 \leq (\|\mathbb{I}_d - sQ\|_2)^k \|x_0 - x^*\|_2$ leads us to the desired estimate for $(\|x_k^s - x^*\|_2)_{k \in \mathbb{N} \cup \{0\}}$.

Now let $t, s \in [c_0, \frac{2}{\mu+L}]$ be given, by using the fact that $\frac{d}{ds}x_k^s = k(\mathbb{I}_d - sQ)^{k-1}Q(x_0 - x^*)$ for all $s > 0$, we can deduce from the mean value theorem that

$$\begin{aligned}
\|x_k^s - x_k^t\|_2 &\leq \left(\sup_{r \in (c_0, \frac{2}{\mu+L})} \|\tfrac{d}{dr}x_k^r\|_2\right)|t - s| \\
&\leq \left(\sup_{r \in (c_0, \frac{2}{\mu+L})} k(\|\mathbb{I}_d - rQ\|_2)^{k-1}\|Q\|_2\|x_0 - x^*\|_2\right)|t - s| \\
&\leq k\left(\sup_{r \in [c_0, \frac{2}{\mu+L}]} \|\mathbb{I}_d - rQ\|_2\right)^{k-1} L|t - s|\|x_0 - x^*\|_2,
\end{aligned}$$

which along with Eq. 23 finishes the proof of the desired sensitivity estimate. □

The next theorem shows that Eq. 21 with stepsize $s \in (0, \frac{2}{\mu+L}]$ is Lipschitz stable in terms of the perturbations of $f$. In particular, for a quadratic function $f \in \mathcal{Q}_{\mu,L}$, we shall establish the Lipschitz

stability with respect to the perturbations in the parameters of $f$. For notational simplicity, we assume $x_0 = 0$ as in Section 3, but it is straightforward to extend the results to an arbitrary initial guess $x_0 \in \mathbb{R}^d$.

**Theorem A.2.** *Let $x_0 = 0$, for each $i \in \{1, 2\}$ let $f_i \in \mathcal{Q}_{\mu,L}$ admit the minimizer $x^{*,i} \in \mathbb{R}^d$ and satisfy $\nabla f_i(x) = Q_i x + b_i$ for a symmetric matrix $Q_i \in \mathbb{R}^{d \times d}$ and $b_i \in \mathbb{R}^d$, for each $i \in \{1, 2\}$, $s > 0$ let $(x^s_{k,i})_{k \in \mathbb{N} \cup \{0\}}$ be the iterates generated by Eq. 21 with $f = f_i$ and stepsize $s$, and let $M = \min(\|x^{*,1}\|_2, \|x^{*,2}\|_2)$. Then we have for all $k \in \mathbb{N}$, $c_0 > 0$, $s \in [c_0, \frac{2}{\mu+L}]$ that:*

$$\|x^s_{k,1} - x^s_{k,2}\|_2 \leq \left[\frac{1}{\mu}\left(1 - (1 - s\mu)^k\right) + sk(1 - s\mu)^{k-1}\right] M\|Q_1 - Q_2\|_2$$
$$+ \frac{1}{\mu}\left(1 - (1 - s\mu)^k\right)\|b_1 - b_2\|_2.$$

*Proof.* Let us assume without loss of generality that $\|x^{*,2}\|_2 \leq \|x^{*,1}\|_2$ and $c_0 \leq \frac{2}{\mu+L}$. We write $\delta x_k = x^s_{k,1} - x^s_{k,2}$ for each $k \in \mathbb{N} \cup \{0\}$. Then, by using Eq. 21 and the fact that $\nabla f_1(x) - \nabla f_1(y) = Q_1(x - y)$ for all $x, y \in \mathbb{R}^d$, we can deduce that $\delta x_0 = 0$ and for all $k \in \mathbb{N} \cup \{0\}$ that

$$\delta x_{k+1} = (\mathbb{I}_d - sQ_1)\delta x_k + e_k = \sum_{i=0}^{k}(\mathbb{I}_d - sQ_1)^i e_{k-i},$$

where $e_k = -s(\nabla f_1 - \nabla f_2)(x^s_{k,2})$ for each $k \in \mathbb{N} \cup \{0\}$. Note that it holds for all $k \in \mathbb{N} \cup \{0\}$ that

$$\|e_k\|_2 \leq s\|(\nabla f_1 - \nabla f_2)(x^s_{k,2})\|_2 \leq s\left(\|Q_2 - Q_2\|_2\|x^s_{k,2}\|_2 + \|b_1 - b_2\|_2\right)$$
$$\leq s\left(\|Q_2 - Q_2\|_2(\|x^{*,2}\|_2 + \|x^s_{k,2} - x^{*,2}\|_2) + \|b_1 - b_2\|_2\right)$$
$$\leq s\left(\|Q_2 - Q_2\|_2(\|x^{*,2}\|_2 + (1 - s\mu)^k\|x_0 - x^{*,2}\|_2) + \|b_1 - b_2\|_2\right),$$

where we have applied Theorem A.1 for the last inequality. Thus for each $k \in \mathbb{N}$, we can obtain from Eq. 23 and $x_0 = 0$ that

$$\|\delta x_k\|_2 \leq \sum_{i=0}^{k-1}(\|\mathbb{I}_d - sQ_1\|_2)^i \|e_{k-1-i}\|_2$$
$$\leq \sum_{i=0}^{k-1}(1 - s\mu)^i s\left[(1 + (1 - s\mu)^{k-1-i})\|x^{*,2}\|_2\|Q_2 - Q_2\|_2 + \|b_1 - b_2\|_2\right]$$
$$= \left[\frac{1}{\mu}\left(1 - (1 - s\mu)^k\right) + sk(1 - s\mu)^{k-1}\right]\min(\|x^{*,1}\|_2, \|x^{*,2}\|_2)\|Q_2 - Q_2\|_2$$
$$+ \frac{1}{\mu}\left(1 - (1 - s\mu)^k\right)\|b_1 - b_2\|_2.$$

which leads to the desired conclusion due to the fact that $M = \min(\|x^{*,1}\|_2, \|x^{*,2}\|_2)$. □

We now proceed to investigate similar properties of the NAG algorithm, whose proofs are more involved due to the fact that NAG is a multi-step method.

Recall that for any given $f \in \mathcal{Q}_{\mu,L}$, initial guess $x_0 \in \mathbb{R}^d$ and stepsize $s \geq 0$, the NAG algorithm generates iterates $(x_k, y_k)_{k \in \mathbb{N} \cup \{0\}}$ as follows: $y_0 = x_0$ and for each $k \in \mathbb{N} \cup \{0\}$,

$$x_{k+1} = y_k - s\nabla f(y_k), \quad y_{k+1} = x_{k+1} + \frac{1 - \sqrt{\mu s}}{1 + \sqrt{\mu s}}(x_{k+1} - x_k). \tag{24}$$

Note that $x_{k+1}, y_{k+1}$ are denoted by $\boldsymbol{y}_{k+1}, \boldsymbol{z}_{k+1}$, respectively, in Section 3.

We first introduce the following matrix $R_{\text{NAG},s}$ for Eq. 24 for any given function $f \in \mathcal{Q}_{\mu,L}$ and stepsize $s \in [0, \frac{4}{3L+\mu}]$:

$$R_{\text{NAG},s} := \begin{pmatrix} (1 + \beta_s)(\mathbb{I}_d - sQ) & -\beta_s(\mathbb{I}_d - sQ) \\ \mathbb{I}_d & 0 \end{pmatrix} \tag{25}$$

where $\beta_s = \frac{1-\sqrt{\mu s}}{1+\sqrt{\mu s}}$ and $Q$ is the Hessian matrix of $f$. The following lemma establishes an upper bound of the spectral norm of the $k$-th power of $R_{\text{NAG},s}$, which extends [18, Lemma 22] to block matrices, a wider range of stepsize ($s$ is allowed to be larger than $1/L$) and a momentum parameter $\beta_s$ depending on the stepsize $s$.

**Lemma A.1.** *Let $f \in \mathcal{Q}_{\mu,L}$, $s \in (0, \frac{4}{3L+\mu}]$, $\beta_s = \frac{1-\sqrt{\mu s}}{1+\sqrt{\mu s}}$ and $R_{NAG,s}$ be defined as in Eq. 25. Then we have for all $k \in \mathbb{N}$ that $\|R_{NAG,s}^k\|_2 \leq 2(k+1)(1-\sqrt{\mu s})^k$.*

*Proof.* Let $Q = U\Lambda U^T$ be the eigenvalue decomposition of the Hessian matrix $Q$ of $f$, where $\Lambda$ is a diagonal matrix comprising of the corresponding eigenvalues of $Q$ sorted in ascending order, i.e., $0 < \mu \leq \lambda_1 \leq \ldots \leq \lambda_d \leq L$. Then we have that

$$R_{\text{NAG},s} = \begin{pmatrix} U & 0 \\ 0 & U \end{pmatrix} \begin{pmatrix} (1+\beta_s)(\mathbb{I}_d - s\Lambda) & -\beta_s(\mathbb{I}_d - s\Lambda) \\ \mathbb{I}_d & 0 \end{pmatrix} \begin{pmatrix} U^T & 0 \\ 0 & U^T \end{pmatrix},$$

which together with the facts that any permutation matrix is orthogonal, and the spectral norm of a matrix is invariant under orthogonal transformations, gives us the identity that: for all $k \in \mathbb{N}$,

$$\|R_{\text{NAG},s}^k\|_2 = \left\| \begin{pmatrix} (1+\beta_s)(\mathbb{I}_d - s\Lambda) & -\beta_s(\mathbb{I}_d - s\Lambda) \\ \mathbb{I}_d & 0 \end{pmatrix}^k \right\|_2 = \max_{i=1,\ldots n} \|T_{s,i}^k\|_2, \qquad (26)$$

where $T_{s,i} = \begin{pmatrix} (1+\beta_s)(1-s\lambda_i) & -\beta_s(1-s\lambda_i) \\ 1 & 0 \end{pmatrix}$ for all $i = 1, \ldots, d$.

Now let $s \in (0, \frac{4}{3L+\mu}]$ and $i = 1, \ldots, d$ be fixed. If $1 - s\lambda_i \geq 0$, by using [18, Lemma 22] (with $\alpha = \mu$, $\beta = 1/s$, $h = 1 - s\lambda_i$ and $\kappa = \beta/\alpha = 1/(\mu s)$), we can obtain that

$$\|T_{s,i}^k\|_2 \leq 2(k+1)\left(\frac{1-\sqrt{\mu s}}{1+\sqrt{\mu s}}(1-\mu s)\right)^{k/2} \leq 2(k+1)(1-\sqrt{\mu s})^k.$$

We then discuss the case where $1 - s\lambda_i < 0$. Let us write $T_{s,i}^k = \begin{pmatrix} a_k & b_k \\ c_k & d_k \end{pmatrix}$ for each $k \in \mathbb{N} \cup \{0\}$, then we have for all $k \in \mathbb{N}$ that

$$a_k = (1+\beta_s)(1-s\lambda_i)a_{k-1} - \beta_s(1-s\lambda_i)c_{k-1}, \quad c_k = a_{k-1},$$
$$b_k = (1+\beta_s)(1-s\lambda_i)b_{k-1} - \beta_s(1-s\lambda_i)d_{k-1}, \quad d_k = b_{k-1},$$

with $a_1 = (1+\beta_s)(1-s\lambda_i)$, $b_1 = -\beta_s(1-s\lambda_i)$, $c_1 = 1$ and $d_1 = 0$. Since the conditions $1 - s\lambda_i < 0$ and $s \leq \frac{4}{3L+\mu}$ imply that $\lambda_i > \frac{1}{s} \geq \frac{3L+\mu}{4} \geq \mu$, we see the discriminant of the characteristic polynomial satisfies that

$$\Delta = (1+\beta_s)^2(1-s\lambda_i)^2 - 4\beta_s(1-s\lambda_i) = \frac{4(1-s\lambda_i)}{(1+\sqrt{\mu s})^2}s(\mu - \lambda_i) > 0,$$

which implies that there exist $l_1, l_2, l_3, l_4 \in \mathbb{R}$ such that it holds for all $k \in \mathbb{N} \cup \{0\}$ that $a_k = l_1\tau_+^{k+1} + l_2\tau_-^{k+1}$ and $b_k = l_3\tau_+^{k+1} + l_4\tau_-^{k+1}$, with $\tau_\pm = \frac{(1+\beta_s)(1-s\lambda_i)\pm\sqrt{\Delta}}{2}$, $l_1 = \frac{1}{\tau_+-\tau_-}$, $l_2 = -\frac{1}{\tau_+-\tau_-}$, $l_3 = \frac{-\tau_-}{\tau_+-\tau_-}$ and $l_4 = \frac{\tau_+}{\tau_+-\tau_-}$. Thus, by letting $\rho_i := \max(|\tau_+|, |\tau_-|)$, we have that $|a_k| = |\sum_{j=0}^k \tau_+^{k-j}\tau_-^j| \leq (k+1)\rho_i^k$ and $|b_k| = |(-\tau_+\tau_-)\sum_{j=0}^{k-1}\tau_+^{k-1-j}\tau_-^j| \leq k\rho_i^{k+1}$ for all $k \in \mathbb{N} \cup \{0\}$.

Now we claim that the conditions $1 - s\lambda_i < 0$ and $0 < s \leq \frac{4}{3L+\mu}$ imply the estimate that $\rho_i \leq 1 - \sqrt{\mu s} < 1$. In fact, the inequality $s \leq \frac{4}{3L+\mu}$ gives us that $\mu s \leq \frac{4\mu}{3L+\mu} \leq 1$, which implies that $\beta_s = \frac{1-\sqrt{\mu s}}{1+\sqrt{\mu s}} \geq 0$. Hence we can deduce from $1 - s\lambda_i < 0$ that $\sqrt{\Delta} \geq (1+\beta_s)(s\lambda_i - 1)$ and

$$|\tau_+| \leq |\tau_-| \leq \frac{s\lambda_i - 1 + \sqrt{(s\lambda_i - 1)s(\lambda_i - \mu)}}{1+\sqrt{\mu s}} \leq \frac{sL - 1 + \sqrt{(sL-1)s(L-\mu)}}{1+\sqrt{\mu s}}.$$

Note that $2 - (\mu + L)s \geq 2 - \frac{4(\mu+L)}{3L+\mu} \geq 0$, we see that

$$\rho_i \leq 1 - \sqrt{\mu s} \Longleftarrow |\tau_-| \leq 1 - \sqrt{\mu s} \Longleftarrow sL - 1 + \sqrt{(sL-1)s(L-\mu)} \leq 1 - \mu s$$
$$\Longleftrightarrow (sL-1)s(L-\mu) \leq (2 - (\mu+L)s)^2$$
$$\Longleftrightarrow (us-1)((3L+\mu)s - 4) \geq 0.$$

Therefore, we have that $\max(|a_k|, |b_k|, |c_k|, |d_k|) \le (k+1)(1 - \sqrt{\mu s})^k$, which, along with the relationship between the spectral norm and Frobenius norm, gives us that $\|T_{s,i}^k\|_2 \le \|T_{s,i}^k\|_F \le 2(k+1)(1 - \sqrt{\mu s})^k$, and finishes the proof of the desired estimate for the case with $1 - s\lambda_i < 0$. $\quad\square$

As an important consequence of Lemma A.1, we now obtain the following upper bound of the error $(\|x_k - x^*\|_2)_{k \in \mathbb{N}}$ for any given objective function $f \in \mathcal{Q}_{\mu,L}$ and stepsize $s \in (0, \frac{4}{3L+\mu}]$.

**Theorem A.3.** *Let* $f \in \mathcal{Q}_{\mu,L}$ *admit the minimizer* $x^* \in \mathbb{R}^d$, $x_0 \in \mathbb{R}^d$, $s \in (0, \frac{4}{3L+\mu}]$ *and* $(x_k^s, y_k^s)_{k \in \mathbb{N} \cup \{0\}}$ *be the iterates generated by Eq. 24 with stepsize* $s$. *Then we have for all* $k \in \mathbb{N} \cup \{0\}$ *that*

$$\|x_{k+1}^s - x^*\|_2^2 + \|x_k^s - x^*\|_2^2 \le 8(1+k)^2(1 - \sqrt{\mu s})^{2k}\|x_0 - x^*\|_2^2.$$

*Proof.* For any $f \in \mathcal{Q}_{\mu,L}$, and $s \in (0, \frac{4}{3L+\mu}]$, by letting $\beta_s = \frac{1 - \sqrt{\mu s}}{1 + \sqrt{\mu s}}$, we can rewrite Eq. 24 as follows: $x_0^s = x_0$, $x_1^s = x_0 - s\nabla f(x_0)$ and for all $k \in \mathbb{N}$,

$$x_{k+1}^s = (1 + \beta_s)x_k^s - \beta_s x_{k-1} - s\nabla f((1 + \beta_s)x_k^s - \beta_s x_{k-1}), \quad (27)$$

which together with the fact that $\nabla f(x^*) = 0$ shows that

$$\begin{pmatrix} x_{k+1}^s - x^* \\ x_k^s - x^* \end{pmatrix} = R_{\text{NAG},s} \begin{pmatrix} x_k^s - x^* \\ x_{k-1}^s - x^* \end{pmatrix} = R_{\text{NAG},s}^k \begin{pmatrix} x_1^s - x^* \\ x_0^s - x^* \end{pmatrix}$$

where $R_{\text{NAG},s}$ is defined as in Eq. 25. Hence by using $x_1^s = x_0 - s\nabla f(x_0)$ and Theorem A.1, we can obtain that

$$\|x_{k+1}^s - x^*\|_2^2 + \|x_k^s - x^*\|_2^2 \le \|R_{\text{NAG},s}^k\|_2^2(\|x_1^s - x^*\|_2^2 + \|x_0^s - x^*\|_2^2)$$
$$\le \|R_{\text{NAG},s}^k\|_2^2 2\|x_0 - x^*\|_2^2,$$

which together with Lemma A.1 leads to the desired convergence result. $\quad\square$

*Remark* A.1. It is well-known that for a general $\mu$-strongly convex and $L$-smooth objective function $f$, one can employ a Lyapunov argument and establish that the iterates obtained by Eq. 24 with stepsize $s \in [0, \frac{1}{L}]$ satisfy the estimate that $\|x_k - x^*\|_2^2 \le \frac{2L}{\mu}(1 - \sqrt{\mu s})^k\|x_0 - x^*\|_2^2$. Here by taking advantage of the affine structure of $\nabla f$, we have obtained a sharper estimate of the convergence rate for a wider range of stepsize $s \in (0, \frac{4}{3L+\mu}]$.

We also would like to emphasize that the upper bound in Theorem A.3 is tight, in the sense that the additional quadratic dependence on $k$ in the error estimate is inevitable. In fact, one can derive a *closed-form expression* of $R_{\text{NAG},s}^k$ and show that, for an index $i$ such that the eigenvalue $\lambda_i$ is sufficiently close to $\mu$, the squared error for that component is of the magnitude $\mathcal{O}((k\sqrt{\mu s} + 1)^2(1 - \sqrt{\mu s})^{2k})$.

We then proceed to analyze the sensitivity of Eq. 24 with respect to the stepsize. The following theorem shows that the iterates $(x_k, y_k)_{k \in \mathbb{N} \cup \{0\}}$ generated by Eq. 24 depend Lipschitz continuously on the stepsize $s$.

**Theorem A.4.** *Let* $f \in \mathcal{Q}_{\mu,L}$ *admit the minimiser* $x^* \in \mathbb{R}^d$, $x_0 \in \mathbb{R}^d$, *and for each* $s \in (0, \frac{4}{3L+\mu}]$ *let* $(x_k^s, y_k^s)_{k \in \mathbb{N} \cup \{0\}}$ *be the iterates generated by Eq. 24 with stepsize* $s$. *Then we have for all* $k \in \mathbb{N}$, $c_0 > 0$ *and* $t, s \in [c_0, \frac{4}{3L+\mu}]$ *that:*

$$\|x_k^t - x_k^s\|_2 \le \left(2L(1+k) + \frac{4}{3}k(k+1)(k+5)\left(\sqrt{\frac{\mu}{c_0}} + 2L\right)\right)(1 - \sqrt{\mu c_0})^k|t - s|\|x_0 - x^*\|_2.$$

*Proof.* Throughout this proof we assume without loss of generality that $c_0 \le s < t \le \frac{4}{3L+\mu}$. Let $Q$ be the Hessian matrix of $f$, for each $r \in [c_0, \frac{4}{3L+\mu}]$ let $\beta_r = \frac{1 - \sqrt{\mu r}}{1 + \sqrt{\mu r}}$, and for each $k \in \mathbb{N} \cup \{0\}$ let $\delta x_k = x_k^t - x_k^s$. Then we can deduce from Eq. 27 that $\delta x_0 = 0$, $\delta x_1 = -(t-s)\nabla f(x_0)$ and for all $k \in \mathbb{N}$ that

$$x_{k+1}^t - x_{k+1}^s = [(1 + \beta_t)x_k^t - \beta_t x_{k-1}^t - t\nabla f((1 + \beta_t)x_k^t - \beta_t x_{k-1}^t)]$$
$$- [(1 + \beta_s)x_k^s - \beta_s x_{k-1}^s - s\nabla f((1 + \beta_s)x_k^s - \beta_s x_{k-1}^s)],$$

which together with the fact that $\nabla f(x) - \nabla f(y) = Q(x - y)$ for all $x, y \in \mathbb{R}^d$ shows that

$$\begin{pmatrix} \delta x_{k+1} \\ \delta x_k \end{pmatrix} = R_{\text{NAG},t} \begin{pmatrix} \delta x_k \\ \delta x_{k-1} \end{pmatrix} + \begin{pmatrix} e_k \\ 0 \end{pmatrix}$$

with $R_{\text{NAG},t}$ defined as in Eq. 25 and the following residual term

$$e_k := [(1 + \beta_t)x_k^s - \beta_t x_{k-1}^s - t\nabla f((1 + \beta_t)x_k^s - \beta_t x_{k-1}^s)]$$
$$- [(1 + \beta_s)x_k^s - \beta_s x_{k-1}^s - s\nabla f((1 + \beta_s)x_k^s - \beta_s x_{k-1}^s)].$$

Hence we can obtain by induction that: for all $k \in \mathbb{N}$,

$$\begin{pmatrix} \delta x_{k+1} \\ \delta x_k \end{pmatrix} = R_{\text{NAG},t}^k \begin{pmatrix} \delta x_1 \\ \delta x_0 \end{pmatrix} + \sum_{i=0}^{k-1} R_{\text{NAG},t}^i \begin{pmatrix} e_{k-i} \\ 0 \end{pmatrix}. \tag{28}$$

Now the facts that $\nabla f(x^*) = 0$ and $\nabla^2 f \equiv Q$ gives us that

$$e_k = (\beta_t - \beta_s)(x_k^s - x_{k-1}^s) - t\nabla f((1 + \beta_t)x_k^s - \beta_t x_{k-1}^s) + s\nabla f((1 + \beta_s)x_k^s - \beta_s x_{k-1}^s)$$
$$= (\beta_t - \beta_s)\big((x_k^s - x^*) - (x_{k-1}^s - x^*)\big) - tQ\big((1 + \beta_t)(x_k^s - x^*) - \beta_t(x_{k-1}^s - x^*)\big)$$
$$+ sQ\big((1 + \beta_s)(x_k^s - x^*) - \beta_s(x_{k-1}^s - x^*)\big)$$
$$= \big[(\beta_t - \beta_s) - (t + t\beta_t - s - s\beta_s)Q\big](x_k^s - x^*) - \big[(\beta_t - \beta_s) - (t\beta_t - s\beta_s)Q\big](x_{k-1}^s - x^*).$$

Note that one can easily verify that the function $g_1(r) = \beta_r$ is $\sqrt{\mu/c_0}$-Lipschitz on $[c_0, \frac{4}{3L+\mu}]$, and the function $g_2(r) = r\beta_r$ is 1-Lipschitz on $[0, \frac{4}{3L+\mu}]$. Moreover, the fact that $f \in \mathcal{Q}_{\mu,L}$ implies that $\|Q\|_2 \leq L$. Thus we can obtain from Theorem A.3 that

$$\|e_k\|_2 \leq \left(\sqrt{\frac{\mu}{c_0}} + 2L\right)|t - s|\|x_k^s - x^*\|_2 + \left(\sqrt{\frac{\mu}{c_0}} + L\right)|t - s|\|x_{k-1}^s - x^*\|_2$$

$$\leq \left(\sqrt{\frac{\mu}{c_0}} + 2L\right)|t - s|\sqrt{2(\|x_k^s - x^*\|_2^2 + \|x_{k-1}^s - x^*\|_2^2)}$$

$$\leq \left(\sqrt{\frac{\mu}{c_0}} + 2L\right)|t - s|4(1 + k)(1 - \sqrt{\mu s})^k\|x_0 - x^*\|_2.$$

This, along with Eq. 28, Lemma A.1 and $s < t$, gives us that

$$\sqrt{\|\delta x_{k+1}\|_2^2 + \|\delta x_k\|_2^2} \leq \|R_{\text{NAG},t}^k\|_2\|\delta x_1\|_2 + \sum_{i=0}^{k-1} \|R_{\text{NAG},t}^i\|_2\|e_{k-i}\|_2$$

$$\leq 2(1 + k)(1 - \sqrt{\mu t})^k|t - s|L\|x_0 - x^*\|_2$$

$$+ \sum_{i=0}^{k-1} 2(1 + i)(1 - \sqrt{\mu t})^i \left(\sqrt{\frac{\mu}{c_0}} + 2L\right)|t - s|4(1 + k - i)(1 - \sqrt{\mu s})^{k-i}\|x_0 - x^*\|_2$$

$$= \left(2L(1 + k) + \frac{4}{3}k(k + 1)(k + 5)\left(\sqrt{\frac{\mu}{c_0}} + 2L\right)\right)|t - s|(1 - \sqrt{\mu s})^k\|x_0 - x^*\|_2,$$

which finishes the proof of the desired estimate due to the fact that $s \geq c_0$. $\qquad \square$

The next theorem is an an analog of Theorem A.2 for the NAC scheme Eq. 24, which shows that the outputs of Eq. 24 with stepsize $s \in (0, \frac{4}{3L+\mu}]$ is Lipschitz stable with respect to the perturbations of the parameters in $f$.

**Theorem A.5.** *Let $x_0 = 0$, for each $i \in \{1, 2\}$ let $f_i \in \mathcal{Q}_{\mu,L}$ admit the minimizer $x^{*,i} \in \mathbb{R}^d$ and satisfy $\nabla f_i(x) = Q_i x + b_i$ for a symmetric matrix $Q_i \in \mathbb{R}^{d \times d}$ and $b_i \in \mathbb{R}^d$, for each $i \in \{1, 2\}$, $s > 0$ let $(x_{k,i}^s)_{k \in \mathbb{N} \cup \{0\}}$ be the iterates generated by Eq. 24 with $f = f_i$ and stepsize $s$, and let $M = \min(\|x^{*,1}\|_2, \|x^{*,2}\|_2)$. Then we have for all $k \in \mathbb{N}$, $s \in [c_0, \frac{4}{3L+\mu}]$ that:*

$$\|x_{k,1}^s - x_{k,2}^s\|_2 \leq \left[\frac{2}{\mu}\left(1 - (1 - \sqrt{\mu s})^{k-1}\right) + s\frac{8(k-1)k(k+4)}{3}(1 - \sqrt{\mu s})^{k-1}\right]M\|Q_1 - Q_2\|_2$$

$$+ \frac{2}{\mu}\left(1 - (1 - \sqrt{\mu s})^k\right)\|b_1 - b_2\|_2.$$

*Proof.* Let us assume without loss of generality that $\|x^{*,2}\|_2 \leq \|x^{*,1}\|_2$. We first fix an arbitrary $s \in [c_0, \frac{4}{3L+\mu}]$ and write $\delta x_k = x_{k,1}^s - x_{k,2}^s$ for each $k \in \mathbb{N} \cup \{0\}$. Then, by using Eq. 27 and the fact that $\nabla f_1(x) - \nabla f_1(y) = Q_1(x-y)$ for all $x, y \in \mathbb{R}^d$, we can deduce that $\delta x_0 = 0$, $\delta x_1 = -s(\nabla f_1 - \nabla f_2)(x_0)$ and for all $k \in \mathbb{N}$,

$$\begin{pmatrix} \delta x_{k+1} \\ \delta x_k \end{pmatrix} = R_{\text{NAG},s} \begin{pmatrix} \delta x_k \\ \delta x_{k-1} \end{pmatrix} + \begin{pmatrix} e_k \\ 0 \end{pmatrix} = R_{\text{NAG},s}^k \begin{pmatrix} \delta x_1 \\ \delta x_0 \end{pmatrix} + \sum_{j=0}^{k-1} R_{\text{NAG},s}^j \begin{pmatrix} e_{k-j} \\ 0 \end{pmatrix}, \qquad (29)$$

where $R_{\text{NAG},s}$ is defined as in Eq. 25 (with $Q = Q_1$) and the residual term $e_k$ is given by

$$e_k := -s(\nabla f_1 - \nabla f_2)((1+\beta_s)x_{k,2}^s - \beta_s x_{k-1,2}^s) \quad \forall k \in \mathbb{N}.$$

Note that, by using Theorem A.3 and the inequality that $x + y \leq \sqrt{2(x^2+y^2)}$ for all $x, y \in \mathbb{R}$, we have for each $k \in \mathbb{N}$ that

$$\begin{aligned}
\|e_k\|_2 &= s\|(Q_1 - Q_2)((1+\beta_s)x_{k,2}^s - \beta_s x_{k-1,2}^s) + (b_1 - b_2)\|_2 \\
&\leq s\|Q_1 - Q_2\|_2(\|x^{*,2}\|_2 + 2\|x_{k,2}^s - x^{*,2}\|_2 + \|x_{k-1,2}^s - x^{*,2}\|_2) + s\|b_1 - b_2\|_2 \\
&\leq s\|Q_1 - Q_2\|_2(\|x^{*,2}\|_2 + 2\|x_{k,2}^s - x^{*,2}\|_2 + \|x_{k-1,2}^s - x^{*,2}\|_2) + s\|b_1 - b_2\|_2 \\
&\leq s\|Q_1 - Q_2\|_2(\|x^{*,2}\|_2 + 8(1+k)(1-\sqrt{\mu s})^k\|x_0 - x^{*,2}\|_2) + s\|b_1 - b_2\|_2.
\end{aligned}$$

Hence we can obtain from Eq. 29, Lemma A.1 and $x_0 = 0$ that

$$\sqrt{\|\delta x_{k+1}\|_2^2 + \|\delta x_k\|_2^2} \leq 2(k+1)(1-\sqrt{\mu s})^k\|\delta x_1\|_2 + \sum_{j=0}^{k-1} 2(j+1)(1-\sqrt{\mu s})^j\|e_{k-j}\|_2$$

$$\leq 2(k+1)(1-\sqrt{\mu s})^k s\|b_1 - b_2\|_2 + \sum_{j=0}^{k-1} 2(j+1)(1-\sqrt{\mu s})^j \big[s\|b_1 - b_2\|_2$$

$$+ s\|Q_1 - Q_2\|_2(1 + 8(1+k-j)(1-\sqrt{\mu s})^{k-j})\|x^{*,2}\|_2\big]$$

$$\leq 2s\sum_{j=0}^{k}(j+1)(1-\sqrt{\mu s})^j\|b_1 - b_2\|_2 + 2s\sum_{j=0}^{k-1}\big[(j+1)(1-\sqrt{\mu s})^j$$

$$+ 8(j+1)(1+k-j)(1-\sqrt{\mu s})^k\big]\|Q_1 - Q_2\|_2\min(\|x^{*,1}\|_2, \|x^{*,2}\|_2).$$

Let $p = 1-\sqrt{\mu s} \in [0,1)$, then we can easily show for each $k \in \mathbb{N} \cup \{0\}$ that $(1-p)\sum_{j=0}^{k}(j+1)p^j = \sum_{j=0}^{k}p^j - p^{k+1}$, which implies that $\sum_{j=0}^{k}(j+1)(1-\sqrt{\mu s})^j \leq \frac{1-(1-\sqrt{\mu s})^{k+1}}{\mu s}$. Moreover, we have that $\sum_{j=0}^{k-1}(j+1)(1+k-j) = \frac{k(k+1)(k+5)}{6}$ for all $k \in \mathbb{N}$. Thus we can simplify the above estimate and deduce for each $k \in \mathbb{N}$ that

$$\|\delta x_{k+1}\|_2 \leq \frac{2}{\mu}\left(1 - (1-\sqrt{\mu s})^{k+1}\right)\|b_1 - b_2\|_2 + \left[\frac{2}{\mu}\left(1 - (1-\sqrt{\mu s})^k\right)\right.$$

$$\left. + s\frac{8k(k+1)(k+5)}{3}(1-\sqrt{\mu s})^k\right]\|Q_1 - Q_2\|_2\min(\|x^{*,1}\|_2, \|x^{*,2}\|_2).$$

Moreover, the condition that $s \leq \frac{4}{3L+\mu} \leq \frac{1}{\mu}$ implies that $\|\delta x_1\|_2 = s\|b_1 - b_2\|_2 \leq \frac{2}{\mu}\left(1 - (1-\sqrt{\mu s})\right)\|b_1 - b_2\|_2$, which shows that the same upper bound also holds for $\|\delta x_1\|_2$ and finishes the proof of the desired estimate. $\qquad \square$

# B Approximation Ability

**Lemma 4.1. (Faster Convergence ⇒ Better Approximation Ability).** *Assume the problem setting in Sec 2. The approximation ability can be bounded by two terms:*

$$\inf_{\phi \in \Phi, \theta \in \Theta} \sup_{Q^* \in \mathcal{Q}^*} P\ell_{\phi,\theta} \leq \sigma_b \mu^{-2} \underbrace{\inf_{\theta \in \Theta} \sup_{Q^* \in \mathcal{Q}^*} P\|Q_\theta - Q^*\|_F}_{\text{approximation ability of the neural module}} + M \underbrace{\inf_{\phi \in \Phi} Cvg(k,\phi)}_{\text{best convergence}}. \tag{11}$$

*Proof.* For each $\phi \in \Phi, \theta \in \Theta, Q^* \in \mathcal{Q}^*$,

$$\ell_{\phi,\theta}(\boldsymbol{x}, \boldsymbol{b}) = \|\mathtt{Alg}_\phi^k(Q_\theta(\boldsymbol{x}), \boldsymbol{b}) - \mathtt{Opt}(Q^*(\boldsymbol{x}), \boldsymbol{b})\|_2 \tag{30}$$

$$\leq \|\mathtt{Alg}_\phi^k(Q_\theta(\boldsymbol{x}), \boldsymbol{b}) - \mathtt{Opt}(Q_\theta(\boldsymbol{x}), \boldsymbol{b})\|_2 + \|\mathtt{Opt}(Q_\theta(\boldsymbol{x}), \boldsymbol{b}) - \mathtt{Opt}(Q^*(\boldsymbol{x}), \boldsymbol{b})\|_2 \tag{31}$$

$$\leq Cvg(k,\phi)\|\mathtt{Alg}_\phi^0(Q_\theta(\boldsymbol{x}), \boldsymbol{b}) - \mathtt{Opt}(Q^*(\boldsymbol{x}), \boldsymbol{b})\|_2 + \|Q_\theta(\boldsymbol{x})^{-1}\boldsymbol{b} - Q^*(\boldsymbol{x})^{-1}\boldsymbol{b}\|_2 \tag{32}$$

$$\leq Cvg(k,\phi) \cdot M + \|\left(Q_\theta(\boldsymbol{x})^{-1} - Q^*(\boldsymbol{x})^{-1}\right)\boldsymbol{b}\|_2, \tag{33}$$

where in the last inequality we have used the facts that the initialization is assumed to be zero vector, i.e., $\mathtt{Alg}_\phi^0(Q_\theta(\boldsymbol{x}), \boldsymbol{b}) = \boldsymbol{0}$, and that $M \geq \sup_{\boldsymbol{x} \in \mathcal{X}, \boldsymbol{b} \in \mathcal{B}} \mathtt{Opt}(Q^*(\boldsymbol{x}), \boldsymbol{b})$. Note that the independence of $(\boldsymbol{x}, \boldsymbol{b})$ and the fact that $\mathbb{E}\boldsymbol{b}\boldsymbol{b}^\top = \sigma_b^2 I$ imply that

$$\mathbb{E}_{\boldsymbol{b}}\|\left(Q_\theta(\boldsymbol{x})^{-1} - Q^*(\boldsymbol{x})^{-1}\right)\boldsymbol{b}\|_2^2 \tag{34}$$

$$= \mathrm{Tr}\left((Q_\theta(\boldsymbol{x})^{-1} - Q^*(\boldsymbol{x})^{-1})^\top (Q_\theta(\boldsymbol{x})^{-1} - Q^*(\boldsymbol{x})^{-1})\sigma_b^2 I\right) \tag{35}$$

$$= \sigma_b^2 \|Q_\theta(\boldsymbol{x})^{-1} - Q^*(\boldsymbol{x})^{-1}\|_F^2 \tag{36}$$

$$= \sigma_b^2 \|Q_\theta(\boldsymbol{x})^{-1}(Q_\theta(\boldsymbol{x}) - Q^*(\boldsymbol{x}))Q^*(\boldsymbol{x})^{-1}\|_F^2 \tag{37}$$

$$\leq \mu^{-4}\sigma_b^2 \|Q_\theta(\boldsymbol{x}) - Q^*(\boldsymbol{x})\|_F^2 \tag{38}$$

Therefore, we see from Hölder's inequality that

$$\mathbb{E}_{\boldsymbol{b}}\|\left(Q_\theta(\boldsymbol{x})^{-1} - Q^*(\boldsymbol{x})^{-1}\right)\boldsymbol{b}\|_2 \leq \mu^{-2}\sigma_b\|Q_\theta(\boldsymbol{x}) - Q^*(\boldsymbol{x})\|_F. \tag{39}$$

Collecting all the above inequalities, we have

$$P\ell_{\phi,\theta} \leq Cvg(k,\phi) \cdot M + \sigma_b\mu^{-2}P\|Q_\theta - Q^*\|_F. \tag{40}$$

Taking supremum over $Q^*$, we have

$$\sup_{Q^* \in \mathcal{Q}^*} P\ell_{\phi,\theta} \leq Cvg(k,\phi) \cdot M + \sigma_b\mu^{-2}\sup_{Q^* \in \mathcal{Q}^*} P\|Q_\theta - Q^*\|_F. \tag{41}$$

Taking infimum over $\phi$ and $\theta$, we have

$$\inf_{\phi \in \Phi, \theta \in \Theta} \sup_{Q^* \in \mathcal{Q}^*} P\ell_{\phi,\theta} \leq \inf_{\phi \in \Phi} Cvg(k,\phi) \cdot M + \sigma_b\mu^{-2}\inf_{\theta \in \Theta} \sup_{Q^* \in \mathcal{Q}^*} P\|Q_\theta - Q^*\|_F. \tag{42}$$

$\square$

**Lemma 4.2. (Faster Convergence ⇒ Better Representation of $Q^*$).** *Assume the problem setting in Sec 2. Then $\forall \phi \in \Phi, \theta \in \Theta, Q^* \in \mathcal{Q}^*$ it holds true that*

$$P\|Q_\theta - Q^*\|_F^2 \leq \sigma_b^{-2}L^4(\sqrt{P\ell_{\phi,\theta}^2} + M \cdot Cvg(k,\phi))^2. \tag{12}$$

*Proof.* Let us assume without loss of generality that $P\ell_{\phi,\theta}^2 = \epsilon$ for some $\epsilon \geq 0$. For any $\boldsymbol{x} \in \mathcal{X}$, $\boldsymbol{b} \in \mathcal{B}$, we have

$$\ell_{\phi,\theta}(\boldsymbol{x}) \geq \|\mathtt{Opt}\left(Q_\theta(\boldsymbol{x}), \boldsymbol{b}\right) - \mathtt{Opt}\left(Q^*(\boldsymbol{x}), \boldsymbol{b}\right)\|_2 - \|\mathtt{Alg}_\phi^k\left(Q_\theta(\boldsymbol{x}), \boldsymbol{b}\right) - \mathtt{Opt}\left(Q_\theta(\boldsymbol{x}), \boldsymbol{b}\right)\|_2$$

$$\geq \|Q_\theta(\boldsymbol{x})^{-1}\boldsymbol{b} - Q^*(\boldsymbol{x})^{-1}\boldsymbol{b}\|_2 - Cvg(k,\phi)\|\mathtt{Opt}\left(Q_\theta(\boldsymbol{x}), \boldsymbol{b}\right)\|_2 \tag{43}$$

$$\geq \|Q_\theta(\boldsymbol{x})^{-1}\boldsymbol{b} - Q^*(\boldsymbol{x})^{-1}\boldsymbol{b}\|_2 - M \cdot Cvg(k,\phi). \tag{44}$$

Rearranging the terms in the above inequality, we have

$$\|Q_\theta(\boldsymbol{x})^{-1}\boldsymbol{b} - Q^*(\boldsymbol{x})^{-1}\boldsymbol{b}\|_2 \leq \ell_{\phi,\theta}(\boldsymbol{x}) + M \cdot Cvg(k,\phi). \tag{45}$$

By Eq. 37 and the inequality that $\|AB\|_F \le \|A\|_2 \|B\|_F$ for any given $A \in \mathbb{R}^{m \times r}$ and $B \in \mathbb{R}^{r \times n}$, we have that

$$\mathbb{E}_{\boldsymbol{b}} \|Q_\theta(\boldsymbol{x})^{-1}\boldsymbol{b} - Q^*(\boldsymbol{x})^{-1}\boldsymbol{b}\|_2^2 \tag{46}$$

$$= \sigma_b^2 \|Q_\theta(\boldsymbol{x})^{-1}(Q_\theta(\boldsymbol{x}) - Q^*(\boldsymbol{x}))Q^*(\boldsymbol{x})^{-1}\|_F^2 \tag{47}$$

$$\ge \sigma_b^2 \frac{\|Q_\theta(\boldsymbol{x}) - Q^*(\boldsymbol{x})\|_F^2}{\|Q^*(\boldsymbol{x})\|_2^2 \|Q_\theta(\boldsymbol{x})\|_2^2} \tag{48}$$

$$\ge \sigma_b^2 \|Q_\theta(\boldsymbol{x}) - Q^*(\boldsymbol{x})\|_F^2 / L^4, \tag{49}$$

which implies that,

$$\|Q_\theta(\boldsymbol{x}) - Q^*(\boldsymbol{x})\|_F^2 \le \sigma_b^{-2} L^4 \mathbb{E}_{\boldsymbol{b}} \|Q_\theta(\boldsymbol{x})^{-1}\boldsymbol{b} - Q^*(\boldsymbol{x})^{-1}\boldsymbol{b}\|_2^2. \tag{50}$$

Combining it with Eq. 45 and the fact that $(P\ell_{\phi,\theta})^2 \le P\ell_{\phi,\theta}^2$, we have

$$P\|Q_\theta(\boldsymbol{x}) - Q^*(\boldsymbol{x})\|_F^2 \le \sigma_b^{-2} L^4 P(\ell_{\phi,\theta} + M \cdot Cvg(k,\phi))^2 \tag{51}$$

$$= \sigma_b^{-2} L^4 \left( P\ell_{\phi,\theta}^2 + (M \cdot Cvg(k,\phi))^2 + 2(M \cdot Cvg(k,\phi))P\ell_{\phi,\theta} \right) \tag{52}$$

$$\le \sigma_b^{-2} L^4 \left( \varepsilon + (M \cdot Cvg(k,\phi))^2 + 2(M \cdot Cvg(k,\phi))\sqrt{\varepsilon} \right) \tag{53}$$

$$= \sigma_b^{-2} L^4 \left( \sqrt{\varepsilon} + M \cdot Cvg(k,\phi) \right)^2, \tag{54}$$

which completes the proof. $\qquad\square$

## C  Generalization Ability

In this section, we shall prove the following result, which is a refined version of Theorem 5.1.

**Theorem C.1.** *Assume the problem setting in Sec 2 and let $r > 0$. Then for any $t > 0$, with probability at least $1 - e^{-t}$, the empirical Rademacher complexity of $\ell_{\mathcal{F}}^{loc}(r)$ can be bounded by*

$$R_n \ell_{\mathcal{F}}^{loc}(r) \leq \sqrt{2} dn^{-\frac{1}{2}} Stab(k) \left( \sqrt{(\sqrt{r} + MCvg(k))^2 C_1(n) + C_2(n,t,k,r)} + 4 \right)$$
$$+ Sens(k) B_{\Phi},$$

*where*

$$C_1(n) = 216\sigma_b^{-2} L^4 \log \mathcal{N}(n^{-\frac{1}{2}}, \ell_{\mathcal{Q}}, L_2(P_n))$$

$$C_2(n,t,k,r) = \left( \frac{768 B_Q^2 t}{n} + 720 B_Q \mathbb{E} R_n \ell_{\mathcal{Q}}^{loc}(r_q) \right) \log \mathcal{N}(n^{-\frac{1}{2}}, \ell_{\mathcal{Q}}, L_2(P_n)),$$

*$r_q = \sigma_b^{-2} L^4 (\sqrt{r} + MCvg(k))^2$, $\ell_{\mathcal{Q}}^{loc}(r_q) = \{\|Q_\theta - Q^*\|_F : \theta \in \Theta, P\|Q_\theta - Q^*\|_F^2 \leq r_q\}$, $B_Q = 2L\sqrt{d}$, and $B_{\Phi} = \frac{1}{2} \sup_{\phi_1, \phi_2 \in \Phi} \|\phi_1 - \phi_2\|_2$.*

*Furthermore, for any $t > 0$, the expected Rademacher complexity of $\ell_{\mathcal{F}}^{loc}(r)$ can be bounded by*

$$\mathbb{E} R_n \ell_{\mathcal{F}}^{loc}(r) \leq \sqrt{2} dn^{-\frac{1}{2}} Stab(k) \left( \sqrt{(\sqrt{r} + MCvg(k))^2 \overline{C}_1(n) + \overline{C}_2(n,t)} + \overline{C}_3(n,t) + 4 \right)$$
$$+ Sens(k) B_{\Phi},$$

*where*

$$\overline{C}_1(n) = 216\sigma_b^{-2} L^4 \log \mathcal{N}_Q,$$

$$\overline{C}_2(n,t) = \left( 1 + 3 B_Q e^{-t} \sqrt{\log \mathcal{N}_Q} + \frac{45}{\sqrt{n}} B_Q \log \mathcal{N}_Q \right) \frac{2880}{\sqrt{n}} B_Q \log \mathcal{N}_Q + t \frac{768 B_Q^2}{n} \log \mathcal{N}_Q,$$

$$\overline{C}_3(n,t) = 12 B_Q e^{-t} \sqrt{\log \mathcal{N}_Q} + \frac{360}{\sqrt{n}} B_Q \log \mathcal{N}_Q,$$

*and $\mathcal{N}_Q = \mathcal{N}(n^{-\frac{1}{2}}, \ell_{\mathcal{Q}}, L_\infty)$.*

In order to prove Theorem C.1, we first prove the following theorem, which reduces bounding the empirical Rademacher complexity of $\ell_{\mathcal{F}}^{loc}(r)$ to that of $\ell_{\mathcal{Q}}^{loc}(r_q)$, and plays an important role in our complexity analysis.

**Theorem C.2.** *Assume the problem setting in Sec 2. Then it holds for any $r > 0$ that*

$$R_n \ell_{\mathcal{F}}^{loc}(r) \leq \sqrt{2} d\, Stab(k) R_n \ell_{\mathcal{Q}}^{loc}(r_q) + Sens(k) B_{\Phi}, \tag{55}$$

*with $r_q = \sigma_b^{-2} L^4 (\sqrt{r} + MCvg(k))^2$, $\ell_{\mathcal{Q}}^{loc}(r_q) = \{\|Q_\theta - Q^*\|_F : \theta \in \Theta, P\|Q_\theta - Q^*\|_F^2 \leq r_q\}$ and $B_{\Phi} = \frac{1}{2} \sup_{\phi_1, \phi_2 \in \Phi} \|\phi_1 - \phi_2\|_2$.*

*Proof.* Let $k \in \mathbb{N}$ be fixed throughout this proof. We first show that the loss $\ell_{\phi, \theta}$ is $Stab(k)$-Lipschtiz in $Q_\theta$ and $Sens(k)$-Lipschtiz in $\phi$. For any $(\boldsymbol{x}, \boldsymbol{b}) \in \mathcal{X} \times \mathcal{B}$, by using the triangle inequality and the definitions of $Stab(k, \phi')$ and $Sens(k)$, we can obtain the following estimate of the loss:

$$|\ell_{\phi, \theta}(\boldsymbol{x}) - \ell_{\phi', \theta'}(\boldsymbol{x})|$$
$$= |\|\text{Alg}_\phi^k(Q_\theta(\boldsymbol{x}), \boldsymbol{b}) - \text{Opt}(Q^*(\boldsymbol{x}), \boldsymbol{b})\|_2 - \|\text{Alg}_{\phi'}^k(Q_{\theta'}(\boldsymbol{x}), \boldsymbol{b}) - \text{Opt}(Q^*(\boldsymbol{x}), \boldsymbol{b})\|_2|$$
$$\leq \|\text{Alg}_\phi^k(Q_\theta(\boldsymbol{x}), \boldsymbol{b}) - \text{Alg}_{\phi'}^k(Q_{\theta'}(\boldsymbol{x}), \boldsymbol{b})\|_2 \tag{56}$$
$$\leq \|\text{Alg}_{\phi'}^k(Q_\theta(\boldsymbol{x}), \boldsymbol{b}) - \text{Alg}_{\phi'}^k(Q_{\theta'}(\boldsymbol{x}), \boldsymbol{b})\|_2 + \|\text{Alg}_\phi^k(Q_\theta(\boldsymbol{x}), \boldsymbol{b}) - \text{Alg}_{\phi'}^k(Q_\theta(\boldsymbol{x}), \boldsymbol{b})\|_2$$
$$\leq Stab(k, \phi')\|Q_\theta(\boldsymbol{x}) - Q_{\theta'}(\boldsymbol{x})\|_2 + Sens(k)\|\phi - \phi'\|_2$$
$$\leq Stab(k)\|Q_\theta(\boldsymbol{x}) - Q_{\theta'}(\boldsymbol{x})\|_2 + Sens(k)\|\phi - \phi'\|_2.$$

where we write $Stab(k) = \sup_{\phi \in \Phi} Stab(k, \phi)$ for each $k \in \mathbb{N}$.

We then establish a vector contraction inequality, which is a modified version of Corollary 4 in [21] and Lemma 5 in [22]. Note that the empirical Rademacher complexity of $\ell_{\mathcal{F}}^{loc}$ can be written as:

$$R_n \ell_{\mathcal{F}}^{loc}(r) = \frac{1}{n} \mathbb{E}_\sigma \sup_{\phi,\theta} \sum_{i=1}^{n} \sigma_i \ell_{\phi,\theta}(\boldsymbol{x}_i) = \frac{1}{n} \mathbb{E}_{\sigma_{1:n-1}} \mathbb{E}_{\sigma_n} \sup_{\phi,\theta} \sum_{i=1}^{n-1} \sigma_i \ell_{\phi,\theta}(\boldsymbol{x}_i) + \sigma_n \ell_{\phi,\theta}(\boldsymbol{x}_n), \quad (57)$$

where the supremum is taken over the parameter space $\{(\phi,\theta) : \phi \in \Phi, \theta \in \Theta, P\ell_{\phi,\theta}^2 \leq r\}$.

Let $U_{n-1}(\phi,\theta) = \sum_{i=1}^{n-1} \sigma_i \ell_{\phi,\theta}(\boldsymbol{x}_i)$ for each $(\phi,\theta)$. We now assume without loss of generality that the supremum can be attained and let

$$\phi_1, \theta_1 = \arg\sup_{\phi,\theta} \big(U_{n-1}(\phi,\theta) + \ell_{\phi,\theta}(\boldsymbol{x}_n)\big),$$

$$\phi_2, \theta_2 = \arg\sup_{\phi,\theta} \big(U_{n-1}(\phi,\theta) - \ell_{\phi,\theta}(\boldsymbol{x}_n)\big),$$

since otherwise we can consider $(\phi_1,\theta_1)$ and $(\phi_2,\theta_2)$ that are $\epsilon$-close to the suprema for any $\epsilon > 0$ and conclude the same result. Then we can deduce from Eq. 56 that

$$\mathbb{E}_{\sigma_n} \sup_{\phi,\theta} \sum_{i=1}^{n-1} \sigma_i \ell_{\phi,\theta}(\boldsymbol{x}_i) + \sigma_n \ell_{\phi,\theta}(\boldsymbol{x}_n)$$

$$= \frac{1}{2}\big(U_{n-1}(\phi_1,\theta_1) + \ell_{\phi_1,\theta_1}(\boldsymbol{x}_n) + U_{n-1}(\phi_2,\theta_2) - \ell_{\phi_2,\theta_2}(\boldsymbol{x}_n)\big)$$

$$= \frac{1}{2}\big(U_{n-1}(\phi_1,\theta_1) + U_{n-1}(\phi_2,\theta_2) + (\ell_{\phi_1,\theta_1}(\boldsymbol{x}_n) - \ell_{\phi_2,\theta_2}(\boldsymbol{x}_n))\big)$$

$$\leq \frac{1}{2}\big(U_{n-1}(\phi_1,\theta_1) + U_{n-1}(\phi_2,\theta_2)\big) + \frac{1}{2}\big(Stab(k)\|Q_{\theta_1}(\boldsymbol{x}_n) - Q_{\theta_2}(\boldsymbol{x}_n)\|_2 + Sens(k)\|\phi_1 - \phi_2\|_2\big)$$

$$\leq \frac{1}{2}\big(U_{n-1}(\phi_1,\theta_1) + U_{n-1}(\phi_2,\theta_2)\big) + \frac{1}{2}Stab(k)\|Q_{\theta_1}(\boldsymbol{x}_n) - Q_{\theta_2}(\boldsymbol{x}_n)\|_F + Sens(k)B_\Phi,$$

where $B_\Phi = \frac{1}{2}\sup_{\phi_1,\phi_2 \in \Phi} \|\phi_1 - \phi_2\|_2$.

For each $\boldsymbol{x} \in \mathcal{X}$, $\theta \in \Theta$ and $1 \leq j, k \leq d$, let $Q_\theta^{j,k}(\boldsymbol{x})$ be the $j,k$-th entry of the matrix $Q_\theta(\boldsymbol{x})$. The the Khintchine-Kahane inequality (see e.g. [21]) gives us that

$$\mathbb{E}_{\sigma_n} \sup_{\phi,\theta} \sum_{i=1}^{n} \sigma_i \ell_{\phi,\theta}(\boldsymbol{x}_i) \leq \frac{1}{2}\left(U_{n-1}(\phi_1,\theta_1) + U_{n-1}(\phi_2,\theta_2)\right) + Sens(k)B_\Phi \qquad (58)$$

$$+ \frac{1}{2}Stab(k)\sqrt{2}\mathbb{E}_{\boldsymbol{\epsilon}_n}\left|\sum_{j,k} \epsilon_n^{j,k}\left(Q_{\theta_1}^{j,k}(\boldsymbol{x}_n) - Q_{\theta_2}^{j,k}(\boldsymbol{x}_n)\right)\right|, \qquad (59)$$

where $\boldsymbol{\epsilon}_n = (\epsilon_n^{j,k})_{j,k=1}^{n}$ are independent Rademacher variables. Hence, if we denote by $s(\boldsymbol{\epsilon}_n)$ the sign of $\sum_{j,k} \epsilon_n^{j,k}\left(Q_{\theta_1}^{j,k}(\boldsymbol{x}_n) - Q_{\theta_2}^{j,k}(\boldsymbol{x}_n)\right)$ and by $Q^{*j,k}(\boldsymbol{x})$ the $j,k$-th entry of the matrix $Q^*(\boldsymbol{x})$, then we can obtain that

$$\mathbb{E}_{\sigma_n} \sup_{\phi,\theta} \sum_{i=1}^{n} \sigma_i \ell_{\phi,\theta}(\boldsymbol{x}_i)$$

$$\leq \mathbb{E}_{\boldsymbol{\epsilon}_n} \frac{1}{2}\left[\left(U_{n-1}(\phi_1,\theta_1) + Stab(k)\sqrt{2}s(\boldsymbol{\epsilon}_n)\sum_{j,k} \epsilon_n^{j,k} Q_{\theta_1}^{j,k}(\boldsymbol{x}_n)\right)\right.$$

$$\left. + \left(U_{n-1}(\phi_2,\theta_2) - Stab(k)\sqrt{2}s(\boldsymbol{\epsilon}_n)\sum_{j,k} \epsilon_n^{j,k} Q_{\theta_2}^{j,k}(\boldsymbol{x}_n)\right)\right] + Sens(k)B_\Phi$$

$$= \mathbb{E}_{\boldsymbol{\epsilon}_n} \frac{1}{2}\left[\left(U_{n-1}(\phi_1,\theta_1) + Stab(k)\sqrt{2}s(\boldsymbol{\epsilon}_n)\sum_{j,k} \epsilon_n^{j,k}\left(Q_{\theta_1}^{j,k}(\boldsymbol{x}_n) - Q^{*j,k}(\boldsymbol{x}_n)\right)\right)\right.$$

$$\left. + \left(U_{n-1}(\phi_2,\theta_2) - Stab(k)\sqrt{2}s(\boldsymbol{\epsilon}_n)\sum_{j,k} \epsilon_n^{j,k}\left(Q_{\theta_2}^{j,k}(\boldsymbol{x}_n) - Q^{*j,k}(\boldsymbol{x}_n)\right)\right)\right] + Sens(k)B_\Phi.$$

Then by taking the supremum over $(\phi, \theta)$ and using the fact that $\sigma_n$ is an independent Rademacher variable, we can deduce that

$$
\mathbb{E}_{\sigma_n} \sup_{\phi,\theta} \sum_{i=1}^{n} \sigma_i \ell_{\phi,\theta}(\boldsymbol{x}_i)
$$

$$
\leq \mathbb{E}_{\boldsymbol{\epsilon}_n} \frac{1}{2} \left[ \sup_{\phi,\theta} \left( U_{n-1}(\phi,\theta) + Stab(k)\sqrt{2}s(\boldsymbol{\epsilon}_n) \sum_{j,k} \epsilon_n^{j,k} \left( Q_\theta^{j,k}(\boldsymbol{x}_n) - Q^{*j,k}(\boldsymbol{x}_n) \right) \right) \right.
$$

$$
\left. + \sup_{\phi,\theta} \left( U_{n-1}(\phi,\theta) - Stab(k)\sqrt{2}s(\boldsymbol{\epsilon}_n) \sum_{j,k} \epsilon_n^{j,k} \left( Q_\theta^{j,k}(\boldsymbol{x}_n) - Q^{*j,k}(\boldsymbol{x}_n) \right) \right) \right] + Sens(k)B_\Phi
$$

$$
= \mathbb{E}_{\boldsymbol{\epsilon}_n} \mathbb{E}_{\sigma_n} \left[ \sup_{\phi,\theta} \left( U_{n-1}(\phi,\theta) + Stab(k)\sqrt{2}\sigma_n \sum_{j,k} \epsilon_n^{j,k} \left( Q_\theta^{j,k}(\boldsymbol{x}_n) - Q^{*j,k}(\boldsymbol{x}_n) \right) \right) \right] + Sens(k)B_\Phi
$$

$$
= \mathbb{E}_{\boldsymbol{\epsilon}_n} \left[ \sup_{\phi,\theta} \left( U_{n-1}(\phi,\theta) + Stab(k)\sqrt{2} \sum_{j,k} \epsilon_n^{j,k} \left( Q_\theta^{j,k}(\boldsymbol{x}_n) - Q^{*j,k}(\boldsymbol{x}_n) \right) \right) \right] + Sens(k)B_\Phi,
$$

where we have used the fact that $\sum_{j,k} \epsilon_n^{j,k} \left( Q_\theta^{j,k}(\boldsymbol{x}_n) - Q^{*j,k}(\boldsymbol{x}_n) \right)$ is a symmetric random variable in the last line.

By proceeding in the same way for all other $\sigma_{n-1}, \cdots, \sigma_1$, we can obtain the following vector-contraction inequality:

$$
\mathbb{E}_\sigma \sup_{\phi,\theta} \sum_{i=1}^{n} \sigma_i \ell_{\phi,\theta}(\boldsymbol{x}_i) \leq \sqrt{2}Stab(k)\mathbb{E}_{\boldsymbol{\epsilon}_{1:n}} \left[ \sup_\theta \sum_{i=1}^{n} \sum_{j,k} \epsilon_i^{j,k} \left( Q_\theta^{j,k}(\boldsymbol{x}_n) - Q^{*j,k}(\boldsymbol{x}_n) \right) \right] \tag{60}
$$

$$
+ nSens(k)B_\Phi.
$$

The first term on the right-hand side can be bounded by using the Cauchy-Schwarz inequality as follows:

$$
\mathbb{E}_{\boldsymbol{\epsilon}_{1:n}} \left[ \sup_\theta \sum_{i=1}^{n} \sum_{j,k} \epsilon_i^{j,k} \left( Q_\theta^{j,k}(\boldsymbol{x}_n) - Q^{*j,k}(\boldsymbol{x}_n) \right) \right]
$$

$$
= \mathbb{E}_{\sigma_{1:n}} \mathbb{E}_{\boldsymbol{\epsilon}_{1:n}} \left[ \sup_\theta \sum_{i=1}^{n} \sigma_i \sum_{j,k} \epsilon_i^{j,k} \left( Q_\theta^{j,k}(\boldsymbol{x}_n) - Q^{*j,k}(\boldsymbol{x}_n) \right) \right]
$$

$$
\leq \mathbb{E}_{\sigma_{1:n}} \mathbb{E}_{\boldsymbol{\epsilon}_{1:n}} \left[ \sup_\theta \sum_{i=1}^{n} \sigma_i \sqrt{\sum_{j,k} (\epsilon_i^{j,k})^2} \sqrt{\sum_{j,k} |Q_\theta^{j,k}(\boldsymbol{x}_n) - Q^{*j,k}(\boldsymbol{x}_n)|^2} \right] \tag{61}
$$

$$
= \mathbb{E}_{\sigma_{1:n}} \mathbb{E}_{\boldsymbol{\epsilon}_{1:n}} \left[ \sup_\theta \sum_{i=1}^{n} \sigma_i d \| Q_\theta(\boldsymbol{x}_n) - Q^*(\boldsymbol{x}_n) \|_F \right]
$$

$$
= d\mathbb{E}_{\sigma_{1:n}} \left[ \sup_\theta \sum_{i=1}^{n} \sigma_i \| Q_\theta(\boldsymbol{x}_n) - Q^*(\boldsymbol{x}_n) \|_F \right].
$$

Therefore, bounding the Rademacher complexity of $\ell_{\mathcal{F}}^{loc}(r)$ reduces to bounding the Rademacher complexity of the space of functions $\|Q_\theta - Q^*\|_F$. Recall that the supremum is taken over the parameter space where $(\phi, \theta) \in \Phi \times \Theta$ satisfies $P\ell_{\phi,\theta}^2 \leq r$. Note that Lemma 4.2 implies that,

$$
P\|Q_\theta - Q^*\|_F^2 \leq r_q := \sigma_b^{-2}L^4 \left( \sqrt{\varepsilon} + M \cdot Cvg(k,\phi) \right)^2. \tag{62}
$$

Hence, by defining the following function space:

$$
\ell_{\mathcal{Q}}^{loc}(r_q) := \left\{ \|Q_\theta - Q^*\|_F : \theta \in \Theta, P\|Q_\theta - Q^*\|_F^2 \leq r_q \right\}, \tag{63}
$$

we can conclude the desired relationship between $R_n\ell_{\mathcal{F}}^{loc}(r)$ and $R_n\ell_{\mathcal{Q}}^{loc}(r_q)$ from the inequalities Eq. 60 and Eq. 61.

$\square$

With Theorem C.2 in hand, we see that, for each $r > 0$, in order to obtain the upper bounds of $R_n \ell_{\mathcal{F}}^{loc}(r)$ in Theorem C.1, it suffices to estimate $R_n \ell_{\mathcal{Q}}^{loc}(r_q)$, i.e., the Rademacher complexity of the function space $\ell_{\mathcal{Q}}^{loc}(r_q)$.

The following theorem summarizes the estimates for the empirical and expected Rademacher complexity of the local class $\ell_{\mathcal{Q}}^{loc}$, which will be established in Propositions C.1 and C.2, respectively.

Recall that, for any given $\epsilon > 0$, a class of functions $\mathcal{F}$ and pseudometric $\| \cdot \|$, the covering number $\mathcal{N}(\epsilon, \mathcal{F}, \| \cdot \|)$ is defined as the cardinality of the smallest subset $\hat{\mathcal{F}}$ of $\mathcal{F}$ for which every element of $\mathcal{F}$ is within the $\epsilon$-neighbourhood of some element of $\hat{\mathcal{F}}$ with respect to the pseudometric $\| \cdot \|$.

**Theorem C.3.** *Assume the problem setting in Sec 2. Let $r > 0$, $r_q = \sigma_b^{-2} L^4 (\sqrt{r} + MCvg(k))^2$ and $\ell_{\mathcal{Q}}^{loc}(r_q) = \{\|Q_\theta - Q^*\|_F : \theta \in \Theta, P\|Q_\theta - Q^*\|_F^2 \leq r_q\}$. Then for all $t > 0$, we have with probability at least $1 - e^{-t}$ that*

$$R_n \ell_{\mathcal{Q}}^{loc}(r_q) \leq n^{-\frac{1}{2}} \left[ \left( C_1(n)(\sqrt{r} + MCvg(k))^2 + C_2(n, t, k, r) \right)^{\frac{1}{2}} + 4 \right], \tag{64}$$

*where*

$$C_1(n) = 216 \sigma_b^{-2} L^4 \log \mathcal{N}\big(n^{-\frac{1}{2}}, \ell_{\mathcal{Q}}, L_2(P_n)\big),$$

$$C_2(n, t, k, r) = \left( \frac{768 B_Q^2 t}{n} + 720 B_Q \mathbb{E} R_n \ell_{\mathcal{Q}}^{loc}(r_q) \right) \log \mathcal{N}\big(n^{-\frac{1}{2}}, \ell_{\mathcal{Q}}, L_2(P_n)\big),$$

*and $B_Q = 2L\sqrt{d}$.*

*Moreover, for all $t > 0$, we have that*

$$\mathbb{E} R_n \ell_{\mathcal{Q}}^{loc}(r_q) \leq n^{-\frac{1}{2}} \left[ \left( \overline{C}_1(n)(\sqrt{r} + MCvg(k))^2 + \overline{C}_2(n, t) \right)^{\frac{1}{2}} + \overline{C}_3(n, t) + 4 \right], \tag{65}$$

*where*

$$\overline{C}_1(n) = 216 \sigma_b^{-2} L^4 \log \mathcal{N}_Q,$$

$$\overline{C}_2(n, t) = \left( 1 + 3 B_Q e^{-t} \sqrt{\log \mathcal{N}_Q} + \frac{45}{\sqrt{n}} B_Q \log \mathcal{N}_Q \right) \frac{2880}{\sqrt{n}} B_Q \log \mathcal{N}_Q + t \frac{768 B_Q^2}{n} \log \mathcal{N}_Q,$$

$$\overline{C}_3(n, t) = 12 B_Q e^{-t} \sqrt{\log \mathcal{N}_Q} + \frac{360}{\sqrt{n}} B_Q \log \mathcal{N}_Q$$

*and $\mathcal{N}_Q = \mathcal{N}(n^{-\frac{1}{2}}, \ell_{\mathcal{Q}}, L_\infty)$.*

We first establish the estimate for the empirical Rademacher complexity $R_n \ell_{\mathcal{Q}}^{loc}(r_q)$, i.e., Eq. 64 in Theorem C.3.

**Proposition C.1.** *Assume the problem setting in Sec 2. Let $B_Q = \sup_{(\theta, \boldsymbol{x}) \in \Theta \times \mathcal{X}} \|Q_\theta(\boldsymbol{x}) - Q^*(\boldsymbol{x})\|_F$, and for each $r > 0$ let $r_q$ and $\ell_{\mathcal{Q}}^{loc}(r_q)$ be defined as in Theorem C.2. Then we have that*

$$R_n \ell_{\mathcal{Q}}^{loc}(r_q) \leq \frac{4}{\sqrt{n}} \left( 1 + 3 B_Q \sqrt{\log \mathcal{N}\big(\tfrac{1}{\sqrt{n}}, \ell_{\mathcal{Q}}^{loc}(r_q), L_2(P_n)\big)} \right). \tag{66}$$

*Moreover, for all $t > 0$, it holds with probability at least $1 - e^{-t}$ that*

$$R_n \ell_{\mathcal{Q}}^{loc}(r_q) \leq \frac{4}{\sqrt{n}} \left( 1 + 3 C(r_q, t) \sqrt{\log \mathcal{N}\big(\tfrac{1}{\sqrt{n}}, \ell_{\mathcal{Q}}^{loc}(r_q), L_2(P_n)\big)} \right), \tag{67}$$

*with the constant $C(r_q, t) = \left( \frac{3 r_q}{2} + \frac{16 B_Q^2 t}{3n} + 5 B_Q \mathbb{E} R_n \ell_{\mathcal{Q}}^{loc}(r_q) \right)^{1/2}$.*

*Proof.* The classical Dudley's entropy integral bound for the empirical Rademacher complexity gives us that

$$R_n \ell_{\mathcal{Q}}^{loc}(r_q) \leq \inf_{\alpha > 0} \left( 4\alpha + \frac{12}{\sqrt{n}} \int_\alpha^\infty \sqrt{\log \mathcal{N}(\epsilon, \ell_{\mathcal{Q}}^{loc}(r_q), L_2(P_n))} \, d\epsilon \right). \tag{68}$$

Observe that all functions in $\ell_{\mathcal{Q}}^{loc}(r_q)$ take value in $[0, B_Q]$, which implies for all $\epsilon \geq B_Q$ that, $\mathcal{N}(\epsilon, \ell_{\mathcal{Q}}^{loc}(r_q), L_2(P_n)) \leq \mathcal{N}(\epsilon, \ell_{\mathcal{Q}}^{loc}(r_q), L_\infty(P_n)) = 1$ and consequently the integrand in Eq. 68 vanishes on $[B_Q, \infty)$. Hence we have that

$$R_n \ell_{\mathcal{Q}}^{loc}(r_q) \leq \inf_{\alpha > 0} \left( 4\alpha + \frac{12}{\sqrt{n}} \int_\alpha^{B_Q} \sqrt{\log \mathcal{N}(\epsilon, \ell_{\mathcal{Q}}^{loc}(r_q), L_2(P_n))} \, d\epsilon \right)$$

$$\leq \frac{4}{\sqrt{n}} + \frac{12}{\sqrt{n}} \int_{\frac{1}{\sqrt{n}}}^{B_Q} \sqrt{\log \mathcal{N}(\epsilon, \ell_{\mathcal{Q}}^{loc}(r_q), L_2(P_n))} \, d\epsilon$$

$$\leq \frac{4}{\sqrt{n}} + \frac{12}{\sqrt{n}} B_Q \sqrt{\log \mathcal{N}\left( \frac{1}{\sqrt{n}}, \ell_{\mathcal{Q}}^{loc}(r_q), L_2(P_n) \right)},$$

where we used the fact that $\mathcal{N}(\epsilon, \ell_{\mathcal{Q}}^{loc}(r_q), L_2(P_n))$ is decreasing in terms of $\epsilon$ for the last inequality. This proves the estimate Eq. 66.

In order to establish the estimate Eq. 67, we shall bound the empirical error $P_n \|Q_\theta - Q^*\|_F^2$ with high probability. Let us consider the class of functions $\ell_{\mathcal{Q}^2}^{loc}(r_q) = \{\|Q_\theta - Q^*\|_F^2 : \theta \in \Theta, P\|Q_\theta - Q^*\|_F^2 \leq r_q\}$, whose element takes values in $[0, B_Q^2]$. Moreover, we see it holds for all $\|Q_\theta - Q^*\|_F^2 \in \ell_{\mathcal{Q}^2}^{loc}(r_q)$ that $P\|Q_\theta - Q^*\|_F^4 \leq B_Q^2 P\|Q_\theta - Q^*\|_F^2 \leq B_Q^2 r_q$. Hence, by applying Theorem 2.1 in [16] (with $\mathcal{F} = \ell_{\mathcal{Q}^2}^{loc}(r_q)$, $a = 0$, $b = B_Q^2$, $\alpha = 1/4$ and $r = B_Q^2 r_q$) and the Cauchy-Schwarz inequality, we can deduce that, for each $t > 0$, it holds with probability at least $1 - e^{-t}$ that

$$P_n \|Q_\theta - Q^*\|_F^2 \leq P\|Q_\theta - Q^*\|_F^2 + \frac{5}{2} \mathbb{E} R_n \ell_{\mathcal{Q}^2}^{loc}(r_q) + \sqrt{\frac{2 B_Q^2 r_q t}{n}} + B_Q^2 \frac{13t}{3n}$$

$$\leq r_q + \frac{5}{2} \mathbb{E} R_n \ell_{\mathcal{Q}^2}^{loc}(r_q) + \frac{r_q}{2} + \frac{B_Q^2 t}{n} + B_Q^2 \frac{13t}{3n}$$

$$\leq \frac{3 r_q}{2} + 5 B_Q \mathbb{E} R_n \ell_{\mathcal{Q}}^{loc}(r_q) + \frac{16 B_Q^2 t}{3n}.$$

Consequently, we see it holds with probability at least $1 - e^{-t}$ that, $\mathcal{N}(\epsilon, \ell_{\mathcal{Q}}^{loc}(r_q), L_2(P_n)) = 1$ for all $\epsilon \geq C(r_q, t)$, with the constant $C(r_q, t)$ defined as in the statement of Proposition C.1. Substituting this fact into the integral bound Eq. 68 and following the same argument as above, we can conclude Eq. 67 with probability at least $1 - e^{-t}$. $\square$

Now we proceed to prove the estimate of the expected Rademacher complexity $\mathbb{E} R_n \ell_{\mathcal{Q}}^{loc}(r_q)$, i.e., Eq. 65 in Theorem C.3.

**Proposition C.2.** *Assume the same setting as in Proposition C.1. Then it holds for any $r, t > 0$ that*

$$\mathbb{E} R_n \ell_{\mathcal{Q}}^{loc}(r_q) \leq n^{-\frac{1}{2}} \left[ \left( C_1(n, t)(\sqrt{r} + M C v g(k))^2 + C_2(n, t) \right)^{\frac{1}{2}} + C_3(n, t) + 4 \right], \qquad (69)$$

*where $C_1(n, t)$, $C_2(n, t)$ and $C_3(n, t)$ the constants defined as in Eq. 72, Eq. 73 and Eq. 74, respectively.*

*Proof.* Let $r, t > 0$ be fixed throughout this proof. Since it holds for all $\epsilon > 0$ and $n \in \mathbb{N}$ that $\mathcal{N}(\epsilon, \ell_{\mathcal{Q}}^{loc}(r_q), L_2(P_n)) \leq \mathcal{N}(\epsilon, \ell_{\mathcal{Q}}, L_\infty)$, we can deduce from Proposition C.1 that

$$\mathbb{E} R_n \ell_{\mathcal{Q}}^{loc}(r_q) \leq \frac{4}{\sqrt{n}} \left( 1 + 3 \left[ C(r_q, t)(1 - e^{-t}) + B_Q e^{-t} \right] \sqrt{\log \mathcal{N}\left( \frac{1}{\sqrt{n}}, \ell_{\mathcal{Q}}, L_\infty \right)} \right), \qquad (70)$$

with the constants $B_Q$ and $C(r_q, t)$ defined as in the statement of Proposition C.1.

The above estimate gives an implicit upper bound of $\mathbb{E} R_n \ell_{\mathcal{Q}}^{loc}(r_q)$ since $C(r_q, t)$ also involves $\mathbb{E} R_n \ell_{\mathcal{Q}}^{loc}(r_q)$. Now we shall introduce the notation $\mathcal{N}_Q^n = \mathcal{N}(\frac{1}{\sqrt{n}}, \ell_{\mathcal{Q}}, L_\infty)$ and derive an explicit upper bound of $\mathbb{E} R_n \ell_{\mathcal{Q}}^{loc}(r_q)$. By rearranging the terms in Eq. 70 and using the definition of $C(r_q, t)$,

we can obtain that

$$\frac{\sqrt{n}}{4}\mathbb{E}R_n\ell_{\mathcal{Q}}^{loc}(r_q) - 1 - 3B_Qe^{-t}\sqrt{\log\mathcal{N}_Q^n}$$

$$\leq 3(1-e^{-t})\sqrt{\left(\frac{3r_q}{2} + \frac{16B_Q^2t}{3n} + 5B_Q\mathbb{E}R_n\ell_{\mathcal{Q}}^{loc}(r_q)\right)\log\mathcal{N}_Q^n}. \tag{71}$$

We shall assume without loss of generality that $\mathbb{E}R_n\ell_{\mathcal{Q}}^{loc}(r_q) \geq \frac{4}{\sqrt{n}}\left(1 + 3B_Qe^{-t}\sqrt{\log\mathcal{N}_Q^n}\right)$, since otherwise we have a trivial estimate that $\mathbb{E}R_n\ell_{\mathcal{Q}}^{loc}(r_q) \leq 4n^{-\frac{1}{2}}A_1$, with $A_1 = 1 + 3B_Qe^{-t}\sqrt{\log\mathcal{N}_Q^n}$. Then by squaring both sides of Eq. 71 and rearranging the terms, we get that

$$\frac{n}{16}(\mathbb{E}R_n\ell_{\mathcal{Q}}^{loc}(r_q))^2 - \left(\frac{\sqrt{n}}{2}A_1 + 45(1-e^{-t})^2B_Q\log\mathcal{N}_Q^n\right)\mathbb{E}R_n\ell_{\mathcal{Q}}^{loc}(r_q)$$

$$+ A_1^2 - 9(1-e^{-t})^2A_2\log\mathcal{N}_Q^n \leq 0,$$

with the constant $A_2 = \frac{3r_q}{2} + \frac{16B_Q^2t}{3n}$. This implies that

$$\mathbb{E}R_n\ell_{\mathcal{Q}}^{loc}(r_q) \leq \frac{8}{n}\left[\frac{\sqrt{n}A_1}{2} + 45(1-e^{-t})^2B_Q\log\mathcal{N}_Q^n + \left(\left[\frac{\sqrt{n}A_1}{2} + 45(1-e^{-t})^2B_Q\log\mathcal{N}_Q^n\right]^2\right.\right.$$

$$\left.\left. - \frac{n}{4}\left[A_1^2 - 9(1-e^{-t})^2A_2\log\mathcal{N}_Q^n\right]\right)^{\frac{1}{2}}\right]$$

$$= n^{-\frac{1}{2}}\left[4A_1 + \frac{360}{\sqrt{n}}(1-e^{-t})^2B_Q\log\mathcal{N}_Q^n + \left(\left[4A_1 + \frac{360}{\sqrt{n}}(1-e^{-t})^2B_Q\log\mathcal{N}_Q^n\right]^2\right.\right.$$

$$\left.\left. - 16\left[A_1^2 - 9(1-e^{-t})^2A_2\log\mathcal{N}_Q^n\right]\right)^{\frac{1}{2}}\right].$$

Hence, for each $t > 0$, by introducing the following constants

$$C_1(n,t) = 216(1-e^{-t})^2\sigma_b^{-2}L^4\log\mathcal{N}_Q^n, \tag{72}$$

$$C_2(n,t) = \left[4A_1 + \frac{360}{\sqrt{n}}(1-e^{-t})^2B_Q\log\mathcal{N}_Q^n\right]^2 - 16A_1^2 + t(1-e^{-t})^2\frac{768B_Q^2}{n}\log\mathcal{N}_Q^n$$

$$= \left(1 + 3B_Qe^{-t}\sqrt{\log\mathcal{N}_Q^n} + \frac{45}{\sqrt{n}}(1-e^{-t})^2B_Q\log\mathcal{N}_Q^n\right)\frac{2880}{\sqrt{n}}(1-e^{-t})^2B_Q\log\mathcal{N}_Q^n$$

$$+ t(1-e^{-t})^2\frac{768B_Q^2}{n}\log\mathcal{N}_Q^n, \tag{73}$$

$$C_3(n,t) = 12B_Qe^{-t}\sqrt{\log\mathcal{N}_Q^n} + \frac{360}{\sqrt{n}}(1-e^{-t})^2B_Q\log\mathcal{N}_Q^n, \tag{74}$$

with $B_Q = \sup_{(\theta,\boldsymbol{x})\in\Theta\times\mathcal{X}}\|Q_\theta(\boldsymbol{x}) - Q^*(\boldsymbol{x})\|_F \leq 2\sqrt{d}L$ and $\mathcal{N}_Q^n = \mathcal{N}(\frac{1}{\sqrt{n}}, \ell_{\mathcal{Q}}, L_\infty)$, we can deduce that

$$\mathbb{E}R_n\ell_{\mathcal{Q}}^{loc}(r_q) \leq n^{-\frac{1}{2}}\left[\left(C_1(n,t)(\sqrt{r} + MCvg(k))^2 + C_2(n,t)\right)^{\frac{1}{2}} + C_3(n,t) + 4\right].$$

$$\square$$

# D RNN as a Neural Algorithm

We denote by $\texttt{RNN}_\phi^k$ a recurrent neural network that has $k$ unrolled RNN cells and view it as a neural algorithm. It has been proposed in [19] to use RNN to learn an optimization algorithm where the update steps in each iteration are given by the operations in an RNN cell

$$\boldsymbol{y}_{k+1} \leftarrow \texttt{RNNcell}\,(Q, \boldsymbol{b}, \boldsymbol{y}_k) := V\sigma\left(W^L\sigma\left(W^{L-1}\cdots W^2\sigma\left(W_1^1\boldsymbol{y}_k + W_2^1\boldsymbol{g}_k\right)\right)\right). \tag{75}$$

In the above equation, we take a specific example where the $\texttt{RNNcell}$ is a multi-layer perception (MLP) with activations $\sigma = \text{RELU}$ that takes the current iterate $\boldsymbol{y}_k$ and the gradient $\boldsymbol{g}_k = Q\boldsymbol{y}_k + \boldsymbol{b}$ as inputs.

**(I) Stable Region.** First, we show that when the parameters satisfy $c_\phi := \sup_Q \|V\|_2\|W_1^1 + W_2^1 Q\|_2 \prod_{l=2}^L \|W^l\|_2 < 1$, the operations in $\texttt{RNNcell}$ are strictly contractive, i.e., $\|\boldsymbol{y}_{k+1} - \boldsymbol{y}_k\|_2 \leq c_\phi\|\boldsymbol{y}_k - \boldsymbol{y}_{k-1}\|_2$.

*Proof.* By definition,

$$\begin{aligned}
\|\boldsymbol{y}_{k+1} - \boldsymbol{y}_k\|_2 &= \|V\sigma\left(W^L\sigma\left(W^{L-1}\cdots W^2\sigma\left(W_1^1\boldsymbol{y}_k + W_2^1\boldsymbol{g}_k\right)\right)\right) \\
&\quad - V\sigma\left(W^L\sigma\left(W^{L-1}\cdots W^2\sigma\left(W_1^1\boldsymbol{y}_{k-1} + W_2^1\boldsymbol{g}_{k-1}\right)\right)\right)\|_2 \\
&\leq \|V\|_2\|\sigma\left(W^L\sigma\left(W^{L-1}\cdots W^2\sigma\left(W_1^1\boldsymbol{y}_k + W_2^1\boldsymbol{g}_k\right)\right)\right) \\
&\quad - \sigma\left(W^L\sigma\left(W^{L-1}\cdots W^2\sigma\left(W_1^1\boldsymbol{y}_{k-1} + W_2^1\boldsymbol{g}_{k-1}\right)\right)\right)\|_2
\end{aligned}$$

Since the activation function $\sigma = \text{RELU}$ satisfies the inequality that $\|\sigma(\boldsymbol{x}) - \sigma(\boldsymbol{x}')\|_2 \leq \|\boldsymbol{x} - \boldsymbol{x}'\|_2$ for any $\boldsymbol{x}, \boldsymbol{x}'$, we have

$$\begin{aligned}
\|\boldsymbol{y}_{k+1} - \boldsymbol{y}_k\|_2 &\leq \|V\|_2\|W^L\sigma\left(W^{L-1}\cdots W^2\sigma\left(W_1^1\boldsymbol{y}_k + W_2^1\boldsymbol{g}_k\right)\right) \\
&\quad - W^L\sigma\left(W^{L-1}\cdots W^2\sigma\left(W_1^1\boldsymbol{y}_{k-1} + W_2^1\boldsymbol{g}_{k-1}\right)\right)\|_2.
\end{aligned}$$

Similarly, we can obtain

$$\begin{aligned}
\|\boldsymbol{y}_{k+1} - \boldsymbol{y}_k\|_2 \\
&\leq \|V\|_2\|W^L\|_2\cdots\|W^2\|_2\|\left(W_1^1\boldsymbol{y}_k + W_2^1\boldsymbol{g}_k\right) - \left(W_1^1\boldsymbol{y}_{k-1} + W_2^1\boldsymbol{g}_{k-1}\right)\|_2 \\
&= \|V\|_2\|W^L\|_2\cdots\|W^2\|_2\|(W_1^1 + QW_2^1)(\boldsymbol{y}_k - \boldsymbol{y}_{k-1})\|_2 \\
&\leq \|V\|_2\|W^L\|_2\cdots\|W^2\|_2\|W_1^1 + QW_2^1\|_2\|\boldsymbol{y}_k - \boldsymbol{y}_{k-1}\|_2 \\
&\leq c_\phi\|\boldsymbol{y}_k - \boldsymbol{y}_{k-1}\|_2.
\end{aligned}$$

Therefore, if $c_\phi < 1$, then the operation is strictly contractive. $\qquad\square$

**(II) Stability.** We shall show the neural algorithm $\texttt{RNN}_\phi^k$ has a stability constant $Stab(k, \phi) = \mathcal{O}(1 - c_\phi^k)$ (see the definition of stability in Sec 3).

*Proof.* Let us consider two quadratic problems induced by $(Q, \boldsymbol{b})$ and $(Q', \boldsymbol{b}')$, and denote the corresponding outputs of $\texttt{RNN}_\phi^k$ as $\boldsymbol{y}_k = \texttt{RNN}_\phi^k(Q, \boldsymbol{b})$ and $\boldsymbol{y}_k' = \texttt{RNN}_\phi^k(Q', \boldsymbol{b}')$.

Denote $c_\phi^Q = \|V\|_2\|W_1^1 + W_2^1 Q\|_2 \prod_{l=2}^L \|W^l\|_2$, $c_\phi^{Q'} = \|V\|_2\|W_1^1 + W_2^1 Q'\|_2 \prod_{l=2}^L \|W^l\|_2$, and $\hat{c}_\phi := \|V\|_2\|W_2^1\|_2 \prod_{l=2}^L \|W^l\|_2$. First, we see that

$$\begin{aligned}
\|\boldsymbol{y}_k\|_2 &\leq c_\phi^Q\|\boldsymbol{y}_{k-1}\|_2 + \hat{c}_\phi\|\boldsymbol{b}\|_2 \leq (c_\phi^Q)^k\|\boldsymbol{y}_0\|_2 + \hat{c}_\phi\|\boldsymbol{b}\|_2 \sum_{i=1}^k (c_\phi^Q)^{i-1} \\
&= \frac{\hat{c}_\phi\|\boldsymbol{b}\|_2(1 - (c_\phi^Q)^k)}{1 - c_\phi^Q} \leq \frac{\hat{c}_\phi\|\boldsymbol{b}\|_2}{1 - c_\phi^Q}.
\end{aligned} \tag{76}$$

Similar conclusion holds for $\boldsymbol{y}_k'$. Then, by following a similar argument as that for the proof of the stable region, we can deduce from $\boldsymbol{y}_0 = \boldsymbol{y}_0'$ that

$$\|\boldsymbol{y}_k - \boldsymbol{y}_k'\|_2$$
$$\leq \|V\|_2\|W^L\|_2\cdots\|W^2\|_2\|(W_1^1 + W_1^2Q)\boldsymbol{y}_{k-1} - (W_1^1 + W_1^2Q')\boldsymbol{y}_{k-1}' + W_1^2(\boldsymbol{b} - \boldsymbol{b}')\|_2$$
$$\leq \|V\|_2\|W^L\|_2\cdots\|W^2\|_2(\|W_1^1 + W_1^2Q\|_2\|\boldsymbol{y}_{k-1} - \boldsymbol{y}_{k-1}'\|_2 + \|Q - Q'\|_2\|W_1^2\|_2\|\boldsymbol{y}_{k-1}'\|_2$$
$$+ \|W_1^2\|_2\|(\boldsymbol{b} - \boldsymbol{b}')\|_2)$$
$$\leq c_\phi^Q\|\boldsymbol{y}_{k-1} - \boldsymbol{y}_{k-1}'\|_2 + \hat{c}_\phi\|Q - Q'\|_2\frac{\hat{c}_\phi\|\boldsymbol{b}'\|_2}{1 - c_\phi^{Q'}} + \hat{c}_\phi\|\boldsymbol{b} - \boldsymbol{b}'\|_2$$
$$\leq (c_\phi^Q)^k\|\boldsymbol{y}_0 - \boldsymbol{y}_0'\|_2 + \left(\frac{\hat{c}_\phi^2\|\boldsymbol{b}'\|_2}{1 - c_\phi^{Q'}}\|Q - Q'\|_2 + \hat{c}_\phi\|\boldsymbol{b} - \boldsymbol{b}'\|_2\right)\sum_{i=1}^k (c_\phi^Q)^{i-1}$$
$$= \frac{\hat{c}_\phi^2\|\boldsymbol{b}'\|_2}{1 - c_\phi^{Q'}}\frac{1 - (c_\phi^Q)^k}{1 - c_\phi^Q}\|Q - Q'\|_2 + \hat{c}_\phi\frac{1 - (c_\phi^Q)^k}{1 - c_\phi^Q}\|\boldsymbol{b} - \boldsymbol{b}'\|_2.$$

Therefore, the stability constant is of the magnitude $\mathcal{O}(1 - c_\phi^k)$. $\qquad\square$

**(III) Sensitivity.** We now proceed to analyze the sensitivity of the neural algorithm $\text{RNN}_\phi^k$ as defined in Sec 3. Note that the strong non-linearity in the RNN cell and the high-dimensionality of the parameter space significantly complicate the analysis of the Lipschitz dependence of $\text{RNN}_\phi^k$ with respect to its parameter $\phi = \{W_1^1, W_1^1, W^2, \ldots, W^L, V\}$. To simplify our presentation, we shall assume the parameter $\phi$ are constrained in a compact subset $\Phi$ of the stable region, and show the neural algorithm $\text{RNN}_\phi^k$ has a sensitivity $Sens(k) = \mathcal{O}(1 - (\inf_{\phi \in \Phi} c_\phi)^k)$. A rigorous sensitivity analysis of RNN with general weights is out of the scope of this paper.

*Proof.* Let the range of parameters $\Phi$ is a compact subset of the stable region, such that for all $\phi \in \Phi$, $c_\phi := \sup_Q \|V\|_2\|W_1^1 + W_2^1Q\|_2\prod_{l=2}^L \|W^l\|_2 \leq c_0 < 1$ for some constant $c_0$. Let $\phi, \phi' \in \Phi$ be two given sets of parameters. For each $k \in \mathbb{N}$, we denote $\boldsymbol{y}_k = \text{RNN}_\phi^k(Q, \boldsymbol{b})$ and $\boldsymbol{y}_k' = \text{RNN}_{\phi'}^k(Q, \boldsymbol{b})$ the outputs corresponding to the parameters $\phi$ and $\phi'$, respectively. Then we have that

$$\|\boldsymbol{y}_k - \boldsymbol{y}_k'\|_2 = \|\text{RNNcell}_\phi(Q, \boldsymbol{b}, \boldsymbol{y}_{k-1}) - \text{RNNcell}_{\phi'}(Q, \boldsymbol{b}, \boldsymbol{y}_{k-1}')\|_2$$
$$\leq \|\text{RNNcell}_\phi(Q, \boldsymbol{b}, \boldsymbol{y}_{k-1}') - \text{RNNcell}_{\phi'}(Q, \boldsymbol{b}, \boldsymbol{y}_{k-1}')\|_2$$
$$+ \|\text{RNNcell}_\phi(Q, \boldsymbol{b}, \boldsymbol{y}_{k-1}) - \text{RNNcell}_\phi(Q, \boldsymbol{b}, \boldsymbol{y}_{k-1}')\|_2$$
$$\leq \|\text{RNNcell}_\phi(Q, \boldsymbol{b}, \boldsymbol{y}_{k-1}') - \text{RNNcell}_{\phi'}(Q, \boldsymbol{b}, \boldsymbol{y}_{k-1}')\|_2 + c_\phi\|\boldsymbol{y}_{k-1} - \boldsymbol{y}_{k-1}'\|_2$$

If there exists a constant $K$, independent of $k, \phi, \phi'$, such that

$$\|\text{RNNcell}_\phi(Q, \boldsymbol{b}, \boldsymbol{y}_{k-1}') - \text{RNNcell}_{\phi'}(Q, \boldsymbol{b}, \boldsymbol{y}_{k-1}')\|_2 \leq K\|\phi - \phi'\|_2, \tag{77}$$

then we can obtain from $\boldsymbol{y}_0 = \boldsymbol{y}_0'$ that

$$\|\boldsymbol{y}_k - \boldsymbol{y}_k'\|_2 \leq v\|\phi - \phi'\|_2 + c_\phi\|\boldsymbol{y}_{k-1} - \boldsymbol{y}_{k-1}'\|_2$$
$$\leq K\|\phi - \phi'\|_2\sum_{i=1}^k c_\phi^{i-1} = \frac{1 - c_\phi^k}{1 - c_\phi}K\|\phi - \phi'\|_2.$$

The fact that $c_\phi \leq c_0 < 1$ for some constant $c_0$ implies that the magnitude of sensitivity is $\mathcal{O}(1 - (\inf_{\phi \in \Phi} c_\phi)^k)$.

Now it remains to establish the estimate Eq. 77. For each $k \in \mathbb{N}$, $\phi = \{W_1^1, W_1^1, W^2, \ldots, W^L, V\}$ and $l = 2, \cdots, L$, we introduce the notation

$$f_\phi^l := W^l\sigma\left(W^{l-1}\cdots W^2\sigma\left(W_1^1\boldsymbol{y}_k + W_2^1\boldsymbol{g}_k\right)\right), \tag{78}$$

with $f_\phi^1 = W_1^1\boldsymbol{y}_k + W_2^1\boldsymbol{g}_k$. Then we have for each $l = 1, \cdots, L$ that

$$\|f_\phi^l\|_2 \leq \prod_{j=2}^l \|W^j\|_2\left(\|W_1^1 + W_2^1Q\|_2\|\boldsymbol{y}_k\|_2 + \|W_2^1\|_2\|\boldsymbol{b}\|_2\right) = c_l\|\boldsymbol{y}_k\|_2 + \hat{c}_l\|\boldsymbol{b}\|_2, \tag{79}$$

with the constants $c_l := \left(\prod_{j=2}^l \|W^j\|_2\right)\|W_1^1 + W_2^1 Q\|_2$, $\hat{c}_l := \left(\prod_{j=2}^l \|W^j\|_2\right)\|W_2^1\|_2$ for all $l = 1, \ldots, L$. Then by induction, we can see that

$$
\begin{aligned}
\|f_\phi^L - f_{\phi'}^L\|_2 &= \|W^L \sigma(f_\phi^{L-1}) - W'^L \sigma(f_{\phi'}^{L-1})\|_2 \\
&\leq \|W^L - W'^L\|_2 \|f_{\phi'}^{L-1}\|_2 + \|W^L\|_2 \|f_\phi^{L-1} - f_{\phi'}^{L-1}\|_2 \\
&\leq \|W^L - W'^L\|_2 \|f_{\phi'}^{L-1}\|_2 + \|W^L\|_2 \Big( \|W^{L-1} - W'^{L-1}\|_2 \|f_{\phi'}^{L-2}\|_2 \\
&\quad + \|W^{L-1}\|_2 \|f_\phi^{L-2} - f_{\phi'}^{L-2}\|_2 \Big) \\
&\leq \sum_{l=2}^L \left( \prod_{j=l+1}^L \|W^j\|_2 \right) \|W^l - W'^l\|_2 \|f_{\phi'}^{l-1}\|_2 + \left( \prod_{l=2}^L \|W^l\|_2 \right) \|f_\phi^1 - f_{\phi'}^1\|_2.
\end{aligned}
$$

Thus we have that

$$
\begin{aligned}
\|\texttt{RNNcell}_\phi(Q, \boldsymbol{b}, \boldsymbol{y}_k) - \texttt{RNNcell}_{\phi'}(Q, \boldsymbol{b}, \boldsymbol{y}_k)\|_2 &= \|V\sigma(f_\phi^L) - V'\sigma(f_{\phi'}^L)\|_2 \\
&\leq \|V - V'\|_2 \|f_{\phi'}^L\|_2 + \|V\|_2 \|f_\phi^L - f_{\phi'}^L\|_2 \\
&\leq \|V - V'\|_2 \|f_{\phi'}^L\|_2 + \|V\|_2 \Bigg[ \sum_{l=2}^L \left( \prod_{j=l+1}^L \|W^j\|_2 \right) \|W^l - W'^l\|_2 \|f_{\phi'}^{l-1}\|_2 \\
&\quad + \left( \prod_{l=2}^L \|W^l\|_2 \right) \|f_\phi^1 - f_{\phi'}^1\|_2 \Bigg].
\end{aligned}
$$

Furthermore, we see that

$$
\begin{aligned}
\|f_\phi^1 - f_{\phi'}^1\|_2 &= \|(W_1^1 + W_2^1 Q)\boldsymbol{y}_k + W_2^1 \boldsymbol{b} - (W_1'^1 + W_2'^1 Q)\boldsymbol{y}_k + W_2'^1 \boldsymbol{b}\|_2 \\
&\leq \|W_1^1 - W_1'^1 + (W_2^1 - W_2'^1)Q\|_2 \|\boldsymbol{y}_k\|_2 + \|W_2^1 - W_2'^1\|_2 \|\boldsymbol{b}\|_2 \\
&\leq \|W_1^1 - W_1'^1\| \|\boldsymbol{y}_k\|_2 + \|W_2^1 - W_2'^1\| (\|Q\|_2 \|\boldsymbol{y}_k\|_2 + \|\boldsymbol{b}\|_2),
\end{aligned}
$$

from which we can conclude that

$$
\begin{aligned}
&\|\texttt{RNNcell}_\phi(Q, \boldsymbol{b}, \boldsymbol{y}_k) - \texttt{RNNcell}_{\phi'}(Q, \boldsymbol{b}, \boldsymbol{y}_k)\|_2 \\
&\leq \|f_{\phi'}^L\|_2 \|V - V'\|_2 + \sum_{l=2}^L \left[ \|V\|_2 \left( \prod_{j=l+1}^L \|W^j\|_2 \right) \|f_{\phi'}^{l-1}\|_2 \right] \|W^l - W'^l\|_2 \\
&\quad + \|V\|_2 \left( \prod_{l=2}^L \|W^l\|_2 \right) \left[ \|W_1^1 - W_1'^1\| \|\boldsymbol{y}_k\|_2 + \|W_2^1 - W_2'^1\| (\|Q\|_2 \|\boldsymbol{y}_k\|_2 + \|\boldsymbol{b}\|_2) \right].
\end{aligned}
$$

Note that we have assumed that the set of parameters $\Phi$ is a compact subset of the stable region and $(Q, \boldsymbol{b}) \in \mathcal{S}_{\mu,L}^{d \times d} \times \mathcal{B}$ are bounded, which imply that for all $\phi, \phi' \in \Phi$, the corresponding outputs $(\boldsymbol{y}_k)_{k \in \mathbb{N}}$ and $(\boldsymbol{y}_k')_{k \in \mathbb{N}}$ are uniformly bound, and hence $\|f_{\phi'}^l\|_2$ is bounded for all $k$ and $l = 1, \ldots, L$ (see Eq. 79). Consequently, we see there exists a constant $K$ such that Eq. 77 is satisfied. This finishes the proof of the desired sensitivity result. $\qquad\square$

**(IV) Convergence.** For the convergence of $\texttt{RNN}_\phi^k$, we can only give the best case guarantee. It is easy to see that with the following choice of $\phi$, $\texttt{RNN}_\phi^k$ can represent $\texttt{GD}_s^k$:

$$
V = [I, -I], \quad W_1^1 = [I; -I]^\top, \quad W_1^2 = [-sI; sI]^\top, \quad W^l = I \text{ for } l = 2, \cdots, L. \tag{80}
$$

Therefore, for the best case, $\texttt{RNN}_\phi^k$ can converge at least as fast as $\texttt{GD}_s^k$.

# E Experiment Details

Here we state the configuration details of the experiments.

- Convexity and smoothness. They are set to be $\mu = 0.1$ and $L = 1$, respectively.
- Dataset. 10000 pairs of $(\boldsymbol{x}, \boldsymbol{b})$ are generated in the following way: 10000 many $\boldsymbol{x}$ are uniformly sampled from $[-5,5]^{10} \times \mathcal{U}^{5 \times 5}$, where $\mathcal{U}^{5 \times 5}$ denotes the space of all $5 \times 5$ unitary matrices. Each input $\boldsymbol{x}$ actually is a tuple $\boldsymbol{x} = (\boldsymbol{z_x}, U_{\boldsymbol{x}})$ where $\boldsymbol{z_x} \in [-5,5]^{10}$ and $U_{\boldsymbol{x}}$ is unitary. 10000 many $\boldsymbol{b}$ are uniformly sampled from $[-5,5]^5$. These 10000 pairs are viewed as the whole dataset.
- Training set $S_n$. During training, $n$ samples are randomly drawn from these 10000 data points as the training set. The labels of these training samples are given by $\boldsymbol{y} = \texttt{Opt}(Q^*(\boldsymbol{x}), \boldsymbol{b})$.
- More details on $Q^*(\boldsymbol{x})$. As mentioned before, each $\boldsymbol{x}$ is a tuple $\boldsymbol{x} = (\boldsymbol{z_x}, U_{\boldsymbol{x}})$. Then we implement $Q^*(\boldsymbol{x}) = U_{\boldsymbol{x}} \text{diag}([g^*(\boldsymbol{z_x}), \mu, L]) U_{\boldsymbol{x}}^\top$, where $g^*$ is a 2-layer dense neural network with hidden dimension 3, output dimension 3, and with randomly fixed parameters. Note that in the final layer of $g^*$, there is a sigmoid-activation that scales the output to the range $[0,1]$ and then the range is further re-scaled to $[\mu, L]$. Finally, $g^*(\boldsymbol{z_x})$ is concatenated with $[\mu, L]$ to form a 5-dimensional vector with smallest and largest value to be $\mu$ and $L$ respectively. This vector represents the eigenvalues of $Q^*(\boldsymbol{x})$.
- Architecture of $Q_\theta$. $Q_\theta$ has the same form as $Q^*(\boldsymbol{x})$, except that the network $g^*$ in $Q^*$ becomes $g_\theta$ in $Q_\theta$. That is, $Q_\theta(\boldsymbol{x}) = U_{\boldsymbol{x}} \text{diag}([g_\theta(\boldsymbol{z_x}), \mu, L]) U_{\boldsymbol{x}}^\top$. Here $g_\theta$ is also a 2-layer dense neural network with output dimension 3, but the hidden dimension can vary. In the reported results, when we say hidden dimension=0, it means $g_\theta$ is a one-layer network.

For the experiments that compare $\texttt{RNN}_\phi^k$ with $\texttt{GD}_\phi^k$ and $\texttt{NAG}_\phi^k$, they are conducted under the 'learning to learn' scenario, with the following modifications compared to the above setting.

- Dataset. Instead of sampling $(\boldsymbol{x}, \boldsymbol{b})$, here we directly sample the problem pairs $(Q, \boldsymbol{b})$. Similarly, 10000 pairs of $(Q, \boldsymbol{b})$ are sampled uniformly from $\mathcal{S}_{\mu,L}^{10 \times 10} \times [-5,5]^{10}$.
- Architecture of $\texttt{RNN}_\phi^k$. For each cell in $\texttt{RNN}_\phi^k$, it is a 4-layer dense neural network with hidden dimension 20-20-20.

For all experiments, each model has been trained by both ADAM and SGD with learning rate searched over [1e-2,5e-3,1e-3,5e-4,1e-4], and only the best result is reported. Furthermore, error bars are produced by 20 independent instantiations of the experiments. The experiments are mainly run parallelly (since we need to search the best learning rate) on clusters which have 416 nodes where on each node there are 24 Xeon 6226 CPU @ 2.70GHz with 192 GB RAM and 1x512 GB SSD.