[Reviews · NeurIPS 2020]

Review 1

Summary and Contributions: This paper proposes an interesting topic on theoretical understandings of deep architectures, aiming to provide a practical guideline for designing reasoning layers. The problem definition is interesting: How will the algorithm properties of a reasoning layer affect the learning behavior of deep architectures containing such layers?

Strengths: Strength: 1. Analyzing the approximation and generalization abilities of such hybrid deep architectures. 2. The analysis matches nicely with experimental observations under various conditions. 3. The paper is well-organized for settings, properties and ability analysis. The claim is reasonable and easy to understand. The results should fit for NeurIPS community.

Weaknesses: This is a very interesting topic and the authors provide enough assumptions to justify their method. However, as has been pointed out, the setting in this paper is very simple and cannot be comparable to the model we utilize now. Since this is not a simple question to answer, the setting provided might not be able to explain the relationship in the deep architectures. It is understandable this would be one of the first few works on this, but enough evidence with practical network setting is required given the impact of this work.

Correctness: Yes.

Clarity: Yes.

Relation to Prior Work: N/A

Reproducibility: No

Additional Feedback: I highly recommend the authors could apply their theory to one real application at least and justify the validity of their method. --- After rebuttal: Good rebuttal though my concern is still not be addressed. But I increase my score due to good explanation. :-)


Review 2

Summary and Contributions: This paper studies hybrid neural network architectures, in which a standard neural network is combined with a "reasoning" or algorithmic layer for tackling tasks more complicated than those typically encountered in deep learning. The main contributions of the paper lie in providing a theoretical study (potentially the first of its kind) of such networks, focusing on convergence, stability to noise and perturbations, as well as providing very abstracted generalization bounds.

Strengths: The most interesting aspect of this paper is the abstraction with which such a large class of potential hybrid models is dealt with. The problem setting is general enough that it's not easy to come up with architectures that would not fit this scheme. While the results part of the paper starts by revisiting some well known results on convergence of gradient descent and Nesterov's method, the study of the sensitivity to perturbations of the two algorithms seems novel. The main interesting results come in Sections 4 and 5, where the authors present first results showing that faster convergence leads to eventual better approximation. I have found the Theorem 5.1 and the corresponding theorems in the Supp. Mat. to be very interesting, if non-trivial to follow. I found it very interesting that the authors studied both the usual Rademacher complexity (impressively dealing with the very abstract reasoning layer), as well as the local version of it - and pointing out their very different behavior! The surprising aspect of the results was the transition in behavior of the algorithms between the over/under-parametrized regime and the "about-right" parametrization. I found it very surprising, that unlike in typical deep nets, where MORE PARAMETERS = BETTER, the authors show that you need about-right parametrization for both better representations and good generalization in these hybrid architectures. The addition of the experimental validation on synthetic data is very interesting, showing nicely this transition in behavior in the generalization gap.

Weaknesses: The main weakness of this work, which the authors acknowledge, are the rather strong assumptions on lines 122 - 128, requiring among others strong convexity. As such, while the results seem to be extremely broad in their abstracted form, I suspect that they might not strictly apply even to the RNN case discussed in Section 6. However, this is a good first step in studying such architectures. The interesting feature of abstraction of the reasoning modules in this paper can be also seen as a weakness of sorts - it is sometimes too abstract and difficult to get a handle on what is happening exactly. This comes with quite a bit of notation and I have found myself continuously having to jump back and forth to make sure of the notation - perhaps reorganizing some of the definitions into a separate section at the beginning could work better, if it could be done without ruining the paper. Perhaps this is my own lack of expert knowledge on all things Rademacher, but I found myself a little bit overwhelmed with the statement of Thm 5.1 and the proof sketch - particularly when terms like Talagrand's inequality for empirical processes or Dudley entropy integral are thrown around without any explanation.

Correctness: The overall theoretical ideas and the methodology are sound. I have put in a bit of effort into following the proofs of most of the statements in the Supp. Mat. and have found no flaw, but that is not to say that none have slipped past me.

Clarity: Overall: yes. More detailed: As mentioned above, I would perhaps try to put more notation in one place, or at least highlight it more. Additionally, while most of the paper is well written, it is clear that different parts were written by different authors. Particularly, I believe Section 6 could use some more attention in terms of having a uniform style and ironing out some grammar issues.

Relation to Prior Work: Yes. The authors clearly specify when some of the results have been known before and have positioned their work well within the broader context.

Reproducibility: Yes

Additional Feedback: In no particular order: - It might be a good idea to discuss the limitations of the assumptions you make in Section 2. - I'm curious whether your framework is general enough that it also encompasses just splitting a deep net into two parts and calling one part the algorithm one - if that is the case, it would be nice to show what you Rademacher statements reduce to! - I believe you haven't defined c_0 in the Table 1 and beyond anywhere in the paper, but I might be wrong. - Why can you say that as k goes to infinity, the space of Alg shrinks to a single function? ---------------------------------------- After author response ---------------------------------------- The authors' response answers some of my confusions, and while I'm still not sure how appicable this would be to non-convex situations, I keep my rating - this is a very interesting abstract framework. The addition of the simple experiment in the response is a nice touch.


Review 3

Summary and Contributions: The paper studies the properties of deep neural network architectures with reasoning layers from a theoretical perspective. They use energy minimization as a way to represent the general process of reasoning and reference. Upon simplified problem settings and assumptions, the authors have tried addressing a non-trivial problem to establish the relationship between the generalization performance of the deep architectures and the properties in the underlying algorithms.

Strengths: 1. I believe the problem presented in this paper is an important topic and the analyses will definitely provide some good insights for the better understanding of deep neural architectures. It is a huge effort made to theoretically prove the convergence of the algorithms and analyzing the same in the light of the performance of the latent models. 2. My understanding is that the authors have tried to use statistical learning techniques for the analysis of the interplay between the underlying algorithm properties and the performance of deep learning models. To avoid the shortcomings of the standard Rademacher, they have used local Rademacher complexity in conjunction with algorithmic properties makes it interesting to read.

Weaknesses: 1. Some of the mathematical notations seem to give ambiguous views on the problem settings and assumptions. May be adding more clarity to the assumptions and justifying them will give better understanding to readers. Otherwise, one may question about the feasibility of the assumption, especially on real-life problems. 2. The paper limits itself to a class of quadratic optimization problems making it applicable for theoretical proofs, however, it would be nice if they can briefly talk about how their method scales up for a complex scenario. 3. Other than Gradient Descent and Nesterov's accelerated gradient method, it might be interesting to see results with more advanced optimizers. 4. Possibly experiments with a real world use case that can cast to a quadratic optimization problem (e.g image segmentation and contour grouping in computer vision) will give a more convincing view of their findings. 5. Not sure if analysis for RNN (or GCN) in Sec 6 can be generalized to dynamic programming (DP) in general as they both algorithmically align with a class of DP algorithms? y_k+1 can be viewed as the solution to the sub-problem at iteration k+1 and DP update is something like the RNNcell update from y_k.

Correctness: I have not verified the proofs thoroughly. The experiments seem to validate their theoretical claims.

Clarity: The paper reads reasonably well.

Relation to Prior Work: Yes

Reproducibility: Yes

Additional Feedback:


Review 4

Summary and Contributions: This paper theoretically discusses the properties of applying the algorithm layer after the neural feature extractor. In this paper, the algorithm layer is defined as the solver for the convex optimization task. The authors mainly discuss the properties of gradient descent (GD) and Nesterov’s accelerated gradient (NAG) algorithm layers. Finally, the authors present the conclusions about the approximation and generalization abilities of the GD and NAG algorithm layers.

Strengths: The assumption of this paper is reasonable. Given the assumption, the author mathematically connects the convergence rate, stability, and Sensitivity of a particular algorithm layer. The final conclusions about the approximation and generalization ability are clear and easy to understand.

Weaknesses: If the author can test the hypothesis in a realistic dataset, it would make the paper more solid, such as the few-shot image classification (https://arxiv.org/pdf/1904.03758.pdf)

Correctness: Yes

Clarity: Yes

Relation to Prior Work: Yes

Reproducibility: Yes

Additional Feedback: -----after author feedback----------------- I have read the authors' rebuttal and other reviewers' comments. I decide to keep my score.

[Author Response · NeurIPS 2020]

We would like to thank all reviewers for their careful reading, thoughtful comments, and overall
positive assessment! We address the questions raised by reviewers below.

**Real-world application [Reviewer #1, #2, #3, #4].** To show the real world applicability of our
theoretical framework, we consider the **local adaptive image denoising** task, where the noise
levels in different parts of the images can be different. More specifically,

**(i) Dataset.** We split BSD500 dataset (400 images) [1] into a training set (100 images) and a test
set (300 images). Gaussian noises are added to each pixel with noise levels depending on image
local smoothness, making the noise levels on edges lower than non-edge regions. The task is to
restore the original image from the noisy version $X \in [0, 1]^{180 \times 180}$.

**(ii) Architecture.** We designed a hybrid architecture $\text{Alg}_\phi^k (E_\theta(X, \cdot))$ where $\text{Alg}_\phi^k$ is a $k$-step
unrolled minimization algorithm to the $\ell_2$-regularized reconstruction objective $E_\theta(X, Y) :=$
$\frac{1}{2}\|Y + g_\theta(X) - X\|_F^2 + \frac{1}{2}\sum_{i,j} |[f_\theta(X)]_{i,j} Y_{i,j}|^2$, and the residual $g_\theta(X)$ and position-wise reg-
ularization coefficient $f_\theta(X)$ are both DnCNN networks as in [2]. The optimization objective,
$E_\theta(X, Y)$, is quadratic in $Y$, which follows the settings our theory focused on.

**(iii) Generalization gap.** We instantiate the hybrid architecture into different models using GD
and NAG algorithms with different unrolled steps $k$. Each model is trained with 3000 epochs,
and the *generalization gaps* between training and test errors are reported in Fig. 1. The results
also show good consistency with our theory, where stabler algorithm (GD) can generalize better
given *over-/under-*parameterized neural module, and for the *about-right* parameterization case,
the generalization gap behaviors similar to $Stab(k) * Cvg(k)$. We will conduct more experimental
trials and provide figures with smoother curves and error bars in our revised paper.

**(iv) Visualization.** To show that the learned hybrid model has a good performance in this real
application, we include a visualization of the original, noisy, and denoised images.

(a) original image

(b) noisy image

(c) denoised by
$\text{GD}_\phi^{12}(E_\theta(X, \cdot))$

**Generality of problem setting [Reviewer #1, #2, #3, #4].**
We acknowledged that our theoretical analysis is performed
under a simplified problem setting, but we'd like to clarify
a few points to avoid confusion.
• We assume $E_\theta(x, y)$ is quadratic in $y$ but it can depend on
*any way* in the input $x$ (i.e., $Q_\theta$ can be any neural network).
This can cover many real applications. For example, the
above image denoising task, and many other data reconstruc-
tion problems can be cast into quadratic energy minimization.
• Even though in the paper we only present the results for

Figure 1: Generalization gap. **Each $k$ corresponds to a
separately trained model.** *Left* (under-parameterized): $f_\theta$ is
a DnCNN with 3 channels and 2 hidden layers and $g_\theta = 0$.
*Middle* (about-right): both $f_\theta$ and $g_\theta$ have 3 channels and 2
hidden layers. *Right (over-parameterized)*: $f_\theta$ has 20 channels.

using GD and NAG algorithms as the reasoning module, the main theorems which state the relation between the learning
behavior and algorithm properties can be applied to *any optimization algorithm* as long as corresponding properties of
the algorithm are provided. **[Reviewer #3]**.
• Our analysis framework can be extended to more general settings where the neural network module outputs a suitable
strongly convex energy function. In fact the key component of our approximation and generalization analysis is the
Lipschitz stability of the maps between the energy function and the exact minimizer, which can be ensured for general
convex optimization problems if suitable regularization terms are introduced in the energy functions. We aim to analyze
this general setting in future research.

**Other questions.**
• Space shrinks to a single function. **[Reviewer #2]** We sincerely thank Reviewer #2 for his/her very detailed comments.
Regarding why $\{\text{Alg}_\phi^\infty : \phi \in \Phi\}$ contains only a single function, this comes from the convergence guarantee of the
algorithms. That is, when the step size $\phi$ is in the stable region $\Phi$, the optimization error will decrease in each iteration
and gradually decrease to 0 when $k \to \infty$. Therefore, for every step size $\phi \in \Phi$, $\text{Alg}_\phi^\infty = \text{Opt}$, the exact minimizer.
• Generalized to DP? **[Reviewer #3]** Thanks Reviewer #3 for bringing in this interesting question, which is also what
we want to address in our future work. Our analysis for RNN/GNN can potentially be adapted to the case where
RNN/GNN are used to learn problems requiring DP, since RNN/GNN can present those operations in DP. However, as
explained in our theory, one may not able to obtain as a tight bound as GD/NAG due to the difficulty of analyzing RNN.
Furthermore, in the case of DP, the notion of convergence with respect the number of step $k$ is different: the $k$ is a fixed
number of stages needed to run the DP iterations to solve the optimization. This will require more research.

[1] Pablo Arbelaez, Michael Maire, Charless Fowlkes, and Jitendra Malik. Contour detection and hierarchical image segmentation.
*IEEE transactions on pattern analysis and machine intelligence*, 33(5):898–916, 2010.
[2] Kai Zhang, Wangmeng Zuo, Yunjin Chen, Deyu Meng, and Lei Zhang. Beyond a gaussian denoiser: Residual learning of
deep51cnn for image denoising. *IEEE Transactions on Image Processing*, 26(7):3142–3155, 2017.


[Meta-Review · NeurIPS 2020]

This paper analyzes approximation ability and generalization ability of deep learning models with reasoning layers. The analysis connects underlying algorithm property and the performance of the deep learning models. In the learning theory analysis, the local Rademacher complexity technique is utilized to obtain tighter bound, which enables to reveal trade-off corresponding to the number of layers. The theoretical findings are justified from numerical experiments. This paper deals with a new problem setting and gives a nice first step. Although its problem setting is quite simple, it is expected that this kind of study will open up a new direction of researches. The numerical experiments support the theoretical analysis well.